# Cusp-singularity-enhanced Coriolis effect for sensitive chip-scale gyroscopes

Sen Zhang[1], Dingbang Xiao[1], Fei Wang[2✉], Ran Huang[3], Lei Yu[4], Ning Zhou[4], Kaixuan He[4], Xuezhong Wu[1], Franco Nori[3✉], Hui Jing[5,6✉] & Xin Zhou[1,2✉]

Gyroscopes, as fundamental inertial sensors, are crucial for rotation measurements in the consumer electronics, automotive and aerospace industries, with the most widely used kind relying on the Coriolis effect[1–6]. The chip-scale Coriolis vibratory gyroscopes (CVGs) show reduced size, weight and cost[1,2] but have far lower performance than traditional macroscale CVGs[3–6], as the weak intrinsic Coriolis factor sets a fundamental limit on scaling the sensitivity against the inherently louder Brownian noise in microchips compared with the macroscale ones. Here, to overcome this physical limit, we propose and experimentally demonstrate the use of third-order singularities lying within cusp catastrophes in the phase-tracked oscillations of an on-chip CVG to facilitate a cubic-root scaling of the Coriolis-effect-induced frequency modulation. Using this effect, we achieve a three-orders-of-magnitude enhancement in the Coriolis factor, yielding a 253-fold improvement in the signal-to-noise ratio and a 297-fold increase in precision. Moreover, the cusp singularity enables a previously unattainable ultrasensitive phase-modulated sublinear measurement, achieving record signal-to-noise ratio performance for silicon-chip gyroscopes. These findings not only provide revolutionary advancements in gyroscope technologies, by filling the gap in observing and controlling the singularity-enhanced Coriolis effect, but also shed new light on other ultrasensitive sensing applications.

Gyroscopes, essential sensors for measuring rotations in free space without the need for external references, play a key role in navigation and stabilization of all kinds of platforms. The most widely used type of gyroscopes operates on the same principle that governs the flight control of certain biological organisms[7,8]—the Coriolis effect, which refers to the perceived deflection of a moving object in a rotating reference frame. Known as CVGs, they sense angular rotation through Coriolis-force-induced interactions between mechanical (phononic) vibratory modes. Traditional CVGs, such as the hemispherical resonator gyroscopes (HRGs), offer high performance and reliability for applications such as oil drilling, maritime navigation and spacecraft pointing[3–6]. However, their high cost limits their widespread use.

Recently, chip-scale CVGs have been developed for much broader uses, including movement monitoring and stabilization control of consumer electronics, automobiles and more[1,2], owing to their reduced size, weight and cost compared with traditional CVGs. Despite these advantages, chip-scale CVGs still lag behind HRGs in performance, making them suitable only for medium-end or low-end applications. Enhancing chip-scale CVG performance to match the level of the current HRGs, while preserving their miniaturization and affordability, is highly desirable to enable revolutionary technologies such as GPS-denied personal navigation, advanced robotics and microsatellites. Yet, this goal remains elusive owing to substantial challenges in microfabrication errors and, more fundamentally, the Brownian noise that increases

as sensor dimensions shrink, which in turn degrades the signal-to-noise ratio (SNR).

At the heart of this challenge is the limited efficiency of the Coriolis interaction, quantified by the Coriolis factor $\kappa_0$ that measures the proportion of modal mass contributing to the Coriolis effect. This factor is intrinsically determined by vibratory-mode geometry and always constrained to $\kappa_0 \leq 1$ (Supplementary Note 1). Consequently, for small rotation rates $\Omega$, the strength of the Coriolis coupling, $2\kappa_0\Omega$, is weak. The resulting physical modulations (for example, amplitude or frequency changes) are easily blurred by the inherently stronger Brownian noise of microscale resonators in chip-scale CVGs relative to macroscale HRGs. Whether and how the Coriolis effect itself can be enhanced beyond this sensitivity limit imposed by the Coriolis factor, $\kappa_0 \leq 1$, remains an unresolved challenge. Resolving this barrier could enable HRG-level performance in chip-scale CVGs and unlock transformative applications.

Relating to this challenge, recent advances in singularity physics[9–13] provide new possibilities for breaking the limit of classical sensing theory, in which sensing responses are proportional to a perturbation $\epsilon$. Instead, $N$th-order singularities can provide a sublinear response[14–19], $\propto \epsilon^{1/N}$, outperforming classical sensors under small perturbations ($|\epsilon| < 1$). For example, up to 20-fold sensitivity enhancements have been reported in the operation of optical gyroscopes near exceptional-point singularities[20,21], surpassing the intrinsic limits of the Sagnac effect[22].

[1]College of Intelligence Science and Technology, National University of Defense Technology, Changsha, China. [2]School of Microelectronics, Southern University of Science and Technology (SUSTech), Shenzhen, China. [3]Center for Quantum Computing (RQC), RIKEN, Wako, Japan. [4]East China Institute of Photo-Electron IC, Bengbu, China. [5]Institute for Quantum Science and Technology, College of Science, National University of Defense Technology, Changsha, China. [6]Key Laboratory of Low-Dimensional Quantum Structures and Quantum Control of Ministry of Education, Hunan Normal University, Changsha, China. ✉e-mail: wangf@sustech.edu.cn; fnori@riken.jp; jinghui73@foxmail.com; zhoux@sustech.edu.cn

Some singularities were also demonstrated to boost SNR in sensing[11,23–27]. However, given the inherent complexity of the Coriolis coupling in CVGs, exploiting singularity enhancement for CVGs remains an unexplored, elusive question.

Here we address the fundamental sensitivity limit in CVGs by proposing and demonstrating the enhancement of the Coriolis effect within a chip-scale CVG, achieved through the introduction and use of third-order singularities residing within cusp catastrophes in a frequency-modulated (FM) operation. We incorporate further stiffness coupling to create such cusp singularities in oscillation-frequency space facilitated by a phase-tracking control. Near the cusp singularities, we obtain a cubic-root responsivity of the singularity-enhanced Coriolis effect, which achieves a three-orders-of-magnitude increase in the Coriolis factor, a 253-fold improvement in the SNR and a 297-fold enhancement in the precision compared with the standard FM operation, surpassing the performance enhancement factors generated from the known exceptional-point mechanisms. Moreover, the cusp singularity enables a previously unattainable ultrasensitive PM rotational measurement, achieving a bias instability of 0.035° h$^{-1}$ and a notable angle random walk (ARW) of 0.00036° (√h)$^{-1}$, roughly an order of magnitude better than the leading silicon-chip gyroscopes, rivalling advanced HRGs of larger size and cost.

## Concept

The theoretical enhancement of the Coriolis effect by singularities starts with the typical CVG model: a proof mass supported by two orthogonal sets of springs (Fig. 1a). The oscillations of the proof mass along the red or blue springs represent standing-wave (SW) modes 1 and 2, with natural frequencies $\omega_{1,2}$ and equal dissipation rate $\gamma$. An out-of-plane rotation at angular velocity $\Omega$ induces Coriolis forces that couple the two modes, giving the CVG Hamiltonian

$$\mathbf{H} = \begin{bmatrix} \omega_1 & i\kappa_0\Omega \\ -i\kappa_0\Omega & \omega_2 \end{bmatrix}, \tag{1}$$

which underpins all kinds of CVG operation. The innovation in our protocol involves adding an intermodal stiffness coupling[28] $g$, implemented as the green spring in Fig. 1a (Supplementary Note 2), which, cooperating with dissipation, enables cusp singularities.

Initially, we examine a standard CVG without stiffness coupling ($g = 0$). Assuming that the modes are degenerate ($\omega_1 = \omega_2 = \omega$), the Coriolis interaction causes a shift of $\pm\kappa_0\Omega$ in the eigenfrequencies, which is used to measure the angular velocity by means of a quadrature frequency-modulated (QFM) operation[29,30]. In this operation, a sinusoidal reference drive, $F_1$, and its $+\pi/2$ phase-shifted counterpart, $F_2$, are applied to modes 1 and 2, respectively. This induces steady-state displacements $q_{1,2} = |q_{1,2}|\cos(\omega_d t + \theta_{1,2})$, with $\omega_d$ the drive frequency and $\theta_{1,2}$ the phase lags relative to $F_1$. For $g = 0$, the quadrature drive establishes a relative mode phase $\vartheta = \theta_2 - \theta_1 = \pi/2$, generating a clockwise (CW) travelling-wave (TW) mode, depicted by the circular proof-mass orbit in Fig. 1a. By tracking the $\theta_1 = -\pi/2$ contour by means of a phase-locked loop (PLL), we can lock into the resonance of the CW mode, as shown in Fig. 1b. The resulting oscillation frequency, $\omega + \kappa_0\Omega$, changes linearly with respect to the angular velocity, with a scale factor determined by the intrinsic Coriolis factor $\kappa_0$. This QFM operation at $g = 0$ serves as the study baseline, representing the standard Coriolis effect.

Notably, activating $g$ greatly changes the $\theta_1$ landscape (Fig. 1c). At zero rotation ($\Omega = 0$), the oscillation frequency tracked to the $-\pi/2$ contour of $\theta_1$, referred to as phase-tracked (PhT) frequency $\omega_T$, exhibits its two 'pitchfork' bifurcations at the critical coupling levels $g = (\pm\sqrt{2} - 1)\gamma$, in which $\omega_T$ becomes highly nonlinear relative to $\Omega$.

The coupled dynamics is fully described by the complex mode susceptibilities $\chi_{1,2}$. Tracking the mode-1 phase $\theta_1 \equiv \arg(\chi_1)$ to $-\pi/2$ gives the PhT frequency through the condition $\text{Re}(\chi_1) = 0$, which leads to the cubic equation (Supplementary Note 3)

$$\delta\omega^3 + \kappa_0\Omega\delta\omega^2 + \left(\frac{\gamma^2}{4} - \frac{g\gamma}{2} - \frac{g^2}{4} - \kappa_0^2\Omega^2\right)\delta\omega$$
$$-\kappa_0\Omega\left(\frac{\gamma^2}{4} + \frac{g^2}{4} + \kappa_0^2\Omega^2\right) = 0, \tag{2}$$

in which $\delta\omega \equiv \omega_T - \omega$ is the frequency modulation. Plotting $\omega_T$ over the $(g, \Omega)$ space under the constraint $\text{Im}(\chi_1) < 0$ yields a partially folded surface exhibiting two cusp catastrophes[13,31–37], as shown in Fig. 1d. The portion with $\text{Im}(\chi_1) > 0$ (the middle sheet for $g > 0$) is omitted, as it corresponds to the $\theta_1 = \pi/2$ contour and is unattainable for the $\theta_1 = -\pi/2$ PhT control. The boundary defined by $\text{Im}(\chi_1) = 0$ signifies a $\theta_1$ phase singularity[38] (PS$_1$) at $g = \gamma$.

The middle (upper or lower) sheets of the folded surface are unstable (stable) (for stability analysis, see Supplementary Note 4). The stability boundaries constitute the catastrophes, as shown by the dashed black curves in Fig. 1d (Supplementary Note 5). Catastrophes tangentially merge at two nexuses, the signature of order-three singularities[36]. These are referred to as cusp singularities, denoted by $X_1$ and $X_2$, as their mappings onto the $g$–$\Omega$ plane form two cusps located at $(g, \Omega) = [(\sqrt{2} - 1)\gamma, 0]$ and $[(-\sqrt{2} - 1)\gamma, 0]$, respectively.

For $g \neq 0$, the proof-mass orbit is no longer circular but an elliptical hybrid mode, described by the state vector $|\psi\rangle = \cos\frac{\phi}{2}|1\rangle + e^{i\vartheta}\sin\frac{\phi}{2}|2\rangle$, in which $\phi = 2\arctan(|q_2|/|q_1|)$. Projecting $|\psi\rangle$ onto a Poincaré sphere by defining spherical coordinates $S_1 = \sin\phi\cos\vartheta$ (ellipticity), $S_2 = \sin\phi\sin\vartheta$ (chirality) and $S_3 = \cos\phi$ (orientation) enables visualizing the mode hybridization (Fig. 1e and Supplementary Note 6). The QFM operation (that is, $g = 0$) maps to the $(0, 1, 0)$ point on the sphere. By examining the state evolutions following the coloured contours on the $\omega_T$ surface in Fig. 1d, which are illustrated by the same-coloured trajectories on the sphere, we locate the cusp singularities $X_1$ and $X_2$ at $(0, 1/\sqrt{2}, -1/\sqrt{2})$ and $(0, -1/\sqrt{2}, 1/\sqrt{2})$, respectively. The PhT states $|\psi\rangle$ can also be interpreted as hybridizations of two TW modes (Supplementary Note 7).

The cusp singularities $X_{1,2}$, having codimension two, are fully regulated by $g$ and $\Omega$. Crucially, the twisted $\omega_T$ geometry near $X_{1,2}$ exhibits sharp modulations under slight changes of $g$ and $\Omega$. An analysis shows that $X_{1,2}$ enable cubic (square) root modulations of $\omega_T$ in response to variations in $\Omega$ ($g$) (Supplementary Note 8), making $X_{1,2}$ highly suitable for the ultrasensitive detection of $\Omega$. In contrast to the standard Coriolis effect (lower panel of Fig. 1b), $X_{1,2}$ can generate disproportionately large changes in $\omega_T$ (outputs) for small $\Omega$ inputs (lower panels of Fig. 1c), highlighting the singular Coriolis effect.

## Experimental realization of the singularities

To verify the singular Coriolis effect and its potential for performance enhancement, we first test the experimental realization of PhT singularities by implementing the protocol in Fig. 1a to a chip-scale CVG, whose core is an electromechanical silicon disc resonator (Methods and Extended Data Fig. 1). The resonator supports a pair of degenerate six-node SW modes 1 and 2 with a natural frequency $\omega_0 \approx 2\pi \times 40.4$ kHz and dissipation rate $\gamma \approx 2\pi \times 0.36$ Hz. These modes are electrostatically actuated, transduced and tuned. The resonator is mounted on a temperature-controlled angular rate table to precisely manage out-of-plane rotation. A quadrature drive applied to modes 1 and 2 excites a CW whispering gallery TW mode. The stiffness coupling is introduced and controlled by a direct-current (DC) tuning voltage $V_c$ applied on the off-axis capacitive electrodes (Methods).

To examine the $\omega_T$ surface, we characterize the $-\pi/2$ contours using the open-loop frequency responses of $\theta_1$, sweeping the drive frequency $\omega_d$ under varying $g$ and $\Omega$ conditions. We examine the contours at constant values of $V_c \in \{-2, -1, 0, 2, 4\}$ V and $\Omega \in \{-100, 0, 100\}°$ s$^{-1}$, as shown

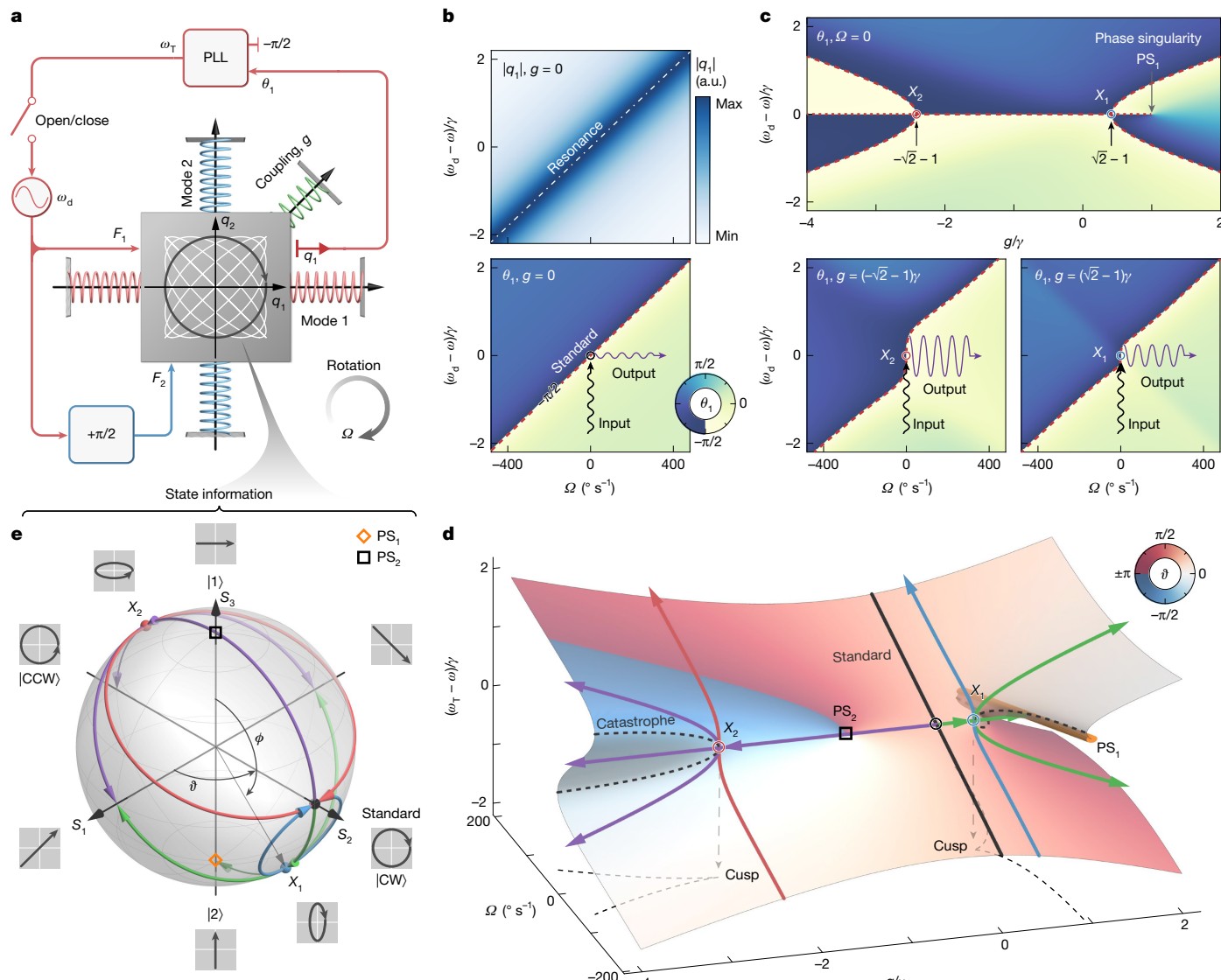

**Fig. 1 | Concept of PhT singularities and enhancement of the Coriolis effect.**
**a**, System schematic: degenerate modes 1 and 2, corresponding to horizontal and vertical oscillations of the proof mass, respectively, are driven by two equal-strength sinusoidal forces in quadrature, $F_{1,2}$, producing a circular mass orbit. A PLL introduced into mode 1 adjusts the oscillation frequency to track $\theta_1$, the phase lag of mode 1 relative to $F_1$, to $-\pi/2$. The Coriolis interaction induced by an out-of-plane rotation $\Omega$ causes a shift in the oscillation frequency. To create cusp singularities, a further coupling, $g$, is incorporated by applying a stiffness perturbation aligned between modes 1 and 2. **b**, For $g = 0$, open-loop responses of mode-1 amplitude $|q_1|$ (upper panel) and phase $\theta_1$ (lower panel) as functions of the drive frequency $\omega_d$ and angular velocity $\Omega$. The resonant frequency (dotted-dashed line) coincides with the $\theta_1 = -\pi/2$ PhT frequency $\omega_T$

(dashed red line), exhibiting a linear response to angular rotation ($\kappa_0\Omega$).
**c**, Phase responses $\theta_1(\omega_d)$ as a function of coupling $g$ for zero angular velocity ($\Omega = 0$, upper panel) and as functions of the angular velocity $\Omega$ under critical coupling conditions $g = (\pm\sqrt{2} - 1)\gamma$ (lower panels), in which the PhT frequency $\omega_T$ exhibits nonlinear responses to $\Omega$. **d**, PhT frequency $\omega_T$ as a function of angular velocity $\Omega$ and coupling strength $g$, showing two cusp catastrophes. The merging points of the catastrophes correspond to cusp singularities $X_{1,2}$, which are associated with two cusps when mapped onto the $g$–$\Omega$ plane. Colour gradient on the surface depicts the relative phase $\vartheta \equiv \theta_2 - \theta_1$. The orange rod represents the $\theta_1$ phase singularity $PS_1$. **e**, Poincaré sphere depicting the PhT state information under various control parameters $g$ and $\Omega$. The coloured trajectories correspond to the contours of the matching colour in **d**. a.u., arbitrary units.

in Fig. 2a,b. The experimental $-\pi/2$ contours of $\theta_1$ (colour edges) match the theoretical model (dotted lines; Supplementary Note 3). Using the fitted parameters from Fig. 2a,b, we restore the PhT frequency $\omega_T$ as a function of $\Omega$ and $V_c$, verifying the existence of the cusp catastrophes (Fig. 2c,d).

Furthermore, we conduct closed-loop experiments with a PLL that tracks the $\theta_1 = -\pi/2$ condition, locking the drive frequency to stable branches of $\omega_T$. We perform bidirectional $\Omega$ sweeps in the range $\pm200°\,s^{-1}$ with $4°\,s^{-1}$ increments, under fixed $V_c \in \{-1.4, -1.2, -1.0, -0.8\}$ V or $\{2.2, 3.0, 4.0, 5.0\}$ V. As shown in Fig. 2c,d, the closed-loop $\omega_T$ data adhere to the stable region of the open-loop-extracted $\omega_T$ surface, verifying the feasibility of the closed-loop readout of $\omega_T$.

Projecting each cusp catastrophe onto the $V_c$–$\Omega$ plane produces two parabolic branches merging at a cusp (Supplementary Note 5), as shown by the black curves in Fig. 2e,f, which are confirmed by the hysteretic transition points of the closed-loop $\omega_T$ data (circles). In the $V_c \leq 0$ case, the catastrophes do not cross $PS_1$ at $V_c \approx -1.852$ V. Through fine-grained $\Omega$ sweeps near the hysteresis thresholds, we locate the two cusp points at $X_1$: ($V_c \approx -0.776$ V, $\Omega \approx -9.1°\,s^{-1}$) and $X_2$: ($V_c \approx 2.170$ V, $\Omega \approx -6.5°\,s^{-1}$). The slight asymmetry of $X_{1,2}$ with respect to $\Omega = 0°\,s^{-1}$ arises from the misalignment error of the off-axis tuning electrodes, causing an unwanted natural frequency mismatch. Readjusting this mismatch repositions the cusp singularities to symmetric locations at $X_1$: ($V_c \approx -0.777$ V, $\Omega \approx 0°\,s^{-1}$) and $X_2$: ($V_c \approx 2.170$ V, $\Omega \approx 0°\,s^{-1}$) (Methods).

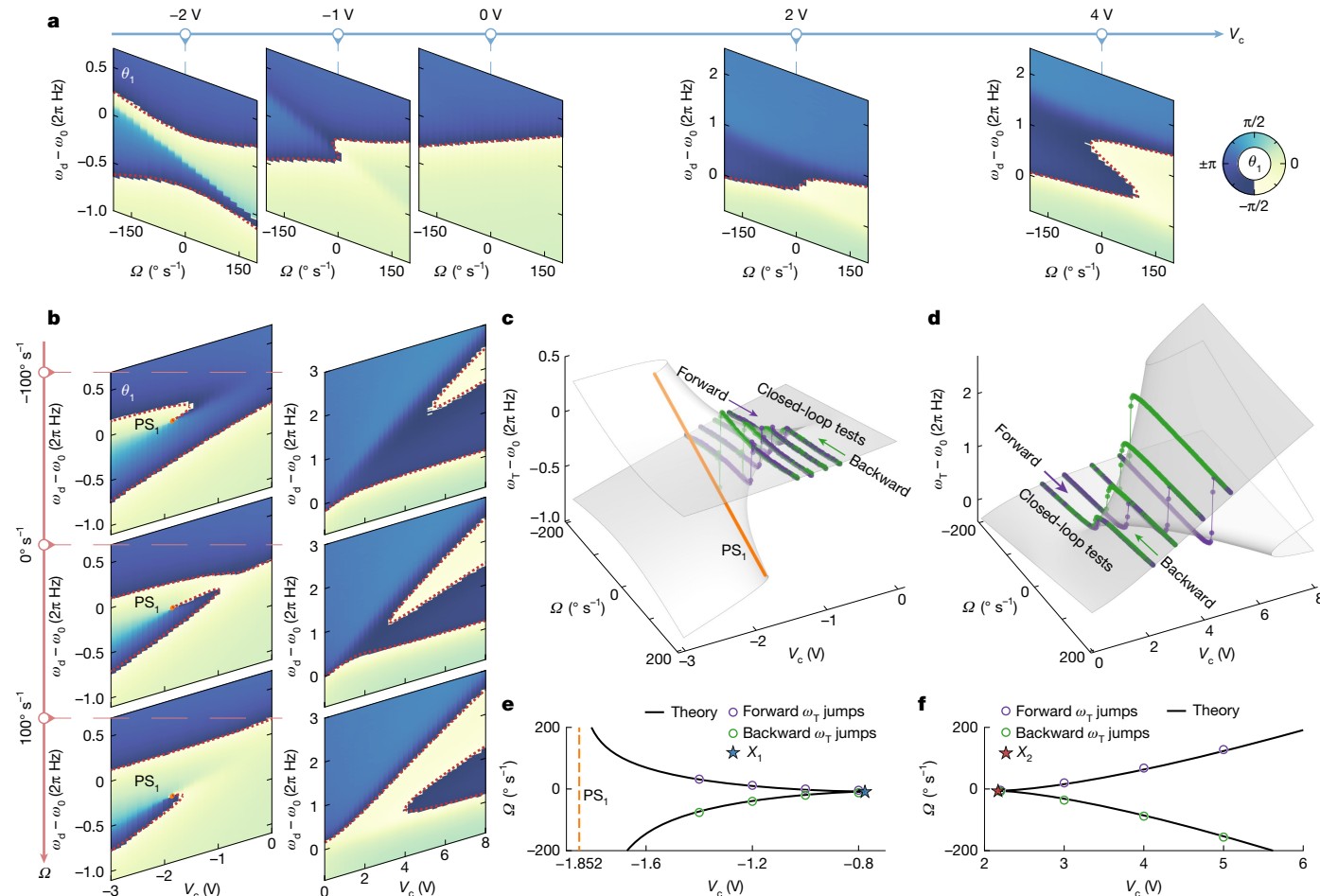

**Fig. 2 | Experimental realization of PhT singularities based on a QFM operation in a chip-scale CVG. a**, Open-loop frequency responses of the mode-1 phase $\theta_1(\omega_d)$ as a function of angular velocity $\Omega$ at constant off-axis tuning voltages: $V_c \in \{-2, -1, 0\}$ V and $\{2, 4\}$ V. In the positive (negative) $V_c$ experiments, two (one) off-axis electrodes are used. The colour edges highlight the $-\pi/2$ contours of $\theta_1$. The dotted curves are the fitting result based on the theoretical model. **b**, Open-loop frequency response of $\theta_1(\omega_d)$ as a function of the off-axis tuning voltage $V_c$ at fixed angular velocities $\Omega \in \{-100, 0, 100\}^{\circ}$ s$^{-1}$ for negative (left panels) and positive (right panels) $V_c$. **c,d**, Surfaces are the theoretically computed PhT frequencies $\omega_T$ as functions of the angular velocity

$\Omega$ and coupling voltage $V_c$ for $V_c \leq 0$ (**c**) and $V_c \geq 0$ (**d**). The orange curve marks the $\theta_1$ phase singularity PS$_1$. The purple (green) points show PLL-enabled closed-loop measurements of $\omega_T$ during forward (backward) angular velocity sweeps from $-200^{\circ}$ s$^{-1}$ to $200^{\circ}$ s$^{-1}$ ($200^{\circ}$ s$^{-1}$ to $-200^{\circ}$ s$^{-1}$) with $4^{\circ}$ s$^{-1}$ increments at fixed coupling voltages of $V_c \in \{-1.4, -1.2, -1.0, -0.8\}$ V (**c**) or $\{2.2, 3.0, 4.0, 5.0\}$ V (**d**). **e,f**, Catastrophic frequency-jumping points (circles) of the closed-loop experiments projected onto the $V_c$–$\Omega$ plane for negative (**e**) and positive (**f**) $V_c$. The solid black curves are theoretical projections of the catastrophes showing cusp-embedded parabolas. The dashed orange line is the projection of the phase singularity PS$_1$.

## Cusp-singularity-enhanced Coriolis effect

Now we demonstrate the enhancement of the Coriolis effect by operating the device at the previously realized PhT cusp singularities $X_{1,2}$. We measure the PhT frequency modulations $\delta\omega_{X1,2}$ as a function of the angular velocity $\Omega$ within a small range $\pm0.15^{\circ}$ s$^{-1}$ with a $0.01^{\circ}$ s$^{-1}$ step. Here each output is compared against a reference recorded at a nearby, constant angular velocity to mitigate errors from the resonant frequency drift (Methods and Extended Data Fig. 2a). The differentially measured $\delta\omega_{X1,2}$ (points in Fig. 3a) agree with the prediction (curves; Supplementary Note 8), showing a sublinear response to $\Omega$, confirming the onset of the singular Coriolis effect. By contrast, the standard Coriolis output $\delta\omega_0$ obtained through $\pm1,000^{\circ}$ s$^{-1}$ QFM measurements, shows a linear sensitivity of $\kappa_0 \approx 0.588$. On a logarithmic scale, the singularity-enhanced outputs have slopes of 1/3 (left side of Fig. 3b), indicating a cubic-root scaling $\delta\omega_{X1,2} \propto \Omega^{1/3}$, whereas the standard Coriolis output maintains a linear dependence (slope of 1).

This scaling change boosts sensitivity, which is characterized by the effective Coriolis factor $\kappa \equiv \delta\omega/\Omega$, as shown in Fig. 3c. Maximal $\kappa$ of 594 and 325 are observed near $X_2$ and $X_1$, respectively, surpassing

the long-standing intrinsic limit $\kappa_0 \leq 1$. Compared with the intrinsic $\kappa_0 \approx 0.588$, a maximal sensitivity amplification factor of 1,010 (553) has been achieved for $X_2$ ($X_1$). Even at faster rotations ($|\Omega| \gg 1^{\circ}$ s$^{-1}$; right side of Fig. 3b), the singularity outputs maintain an advantage over the standard Coriolis output, despite the enhancement factors gradually diminishing with increasing $|\Omega|$ (Fig. 3c). This reconciles the trade-off between high sensitivity at small inputs and a wide dynamic range.

Then, zero-$\Omega$ bias FM outputs at $X_{1,2}$ and in the baseline (QFM, $g = 0$) configuration were recorded over a 10-min duration to calculate the Allan deviations $\sigma_{\delta\omega}$ (Fig. 3d). The bias frequency drifts at $X_2$ ($X_1$) exceed that of the QFM baseline by 1.5 (1.3) times. This degradation, attributed to the singularity's amplification of input errors, is expected to be surpassed by the sensitivity enhancements.

Figure 3e shows the effective input deviations $\sigma_\Omega$, corresponding to the bias frequency deviations $\sigma_{\delta\omega}$, calculated using the theoretical $\omega_T$–$\Omega$ relation (Supplementary Note 10 and Extended Data Fig. 3a). Two key performance metrics are considered: the ARW, reflecting the SNR in the short term, and the bias instability, evaluating precision in the longer term. The $X_2$ ($X_1$) configuration yields a 253-fold (37-fold) decrease in ARW and a 297-fold (53-fold) reduction in bias instability compared

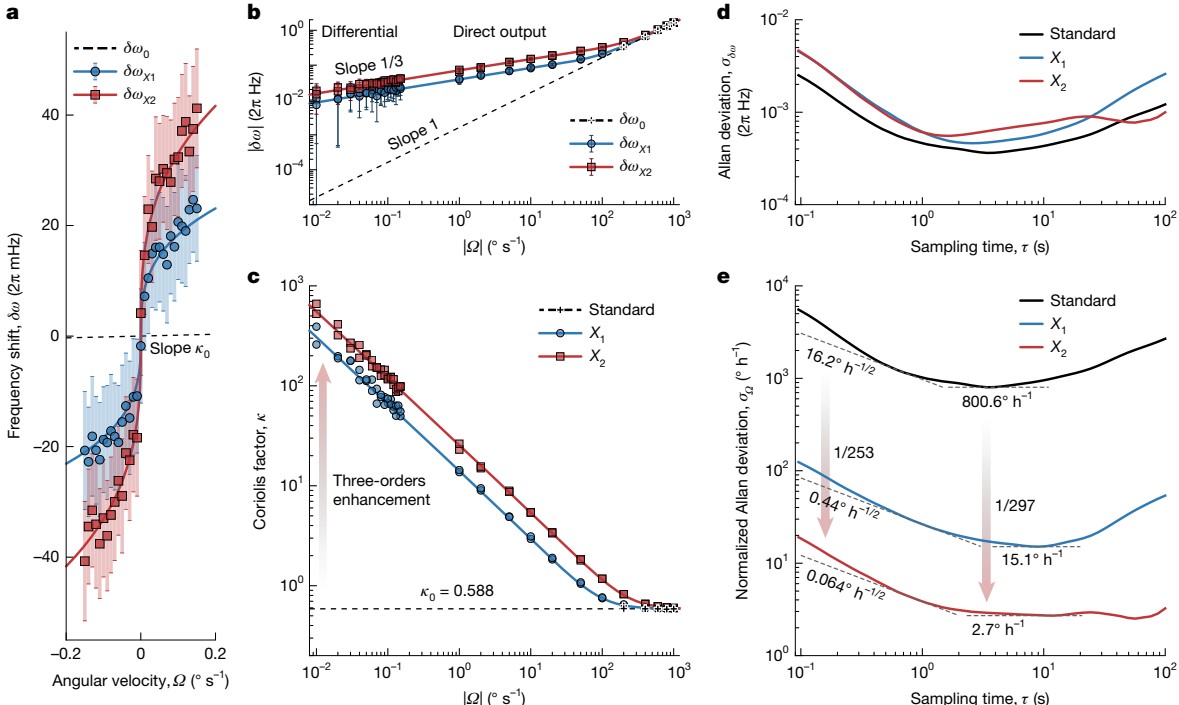

**Fig. 3 | Cusp-singularity-enhanced frequency modulation. a**, Experimentally observed shifts in the PhT frequencies (red and blue points) as a function of small rotation perturbations within ±0.15° s⁻¹, stepping by 0.01° s⁻¹, when operating near the singularities $X_{1,2}$. These data are obtained using differential measurements to mitigate resonant frequency fluctuations. The red and blue curves are theoretical $\omega_T$ shifts for $g = (-\sqrt{2}-1)\gamma$ and $(\sqrt{2}-1)\gamma$, respectively. The dashed black line represents the fitted standard QFM Coriolis output. The error bars represent the standard deviations. **b**, Logarithmic plots of the observed frequency shifts as a function of angular velocity $\Omega$ for both slow and fast rotation measurements. Data are shown for operations initiated at singularities $X_1$ and $X_2$ (red and blue points) and standard QFM mode (black points). The fast-rotation ($|\Omega| > 1°$ s⁻¹) data are measured without using a differential configuration. The red, blue and black curves are theoretical $\omega_T$ for $g = (-\sqrt{2}-1)\gamma$, $(\sqrt{2}-1)\gamma$ and 0, respectively. The singularity-mediated outputs exhibit slopes of 1/3, whereas the standard Coriolis output shows a slope of 1. **c**, Experimental (points) and theoretical (curves) Coriolis factor $\kappa \equiv \delta\omega/\Omega$ describing the sensitivity of the frequency output for singularity-mediated operations and standard operation. For minimal inputs of ±0.01° s⁻¹, the mean values of $\kappa$ are 594 (for $X_2$) and 325 (for $X_1$), representing 1,010-fold and 553-fold enhancements, respectively, compared with the intrinsic value $\kappa_0 \approx 0.588$. **d,e**, Allan deviations of the zero-$\Omega$ bias frequency signals (**d**) and the corresponding effective bias rotations for singularity-enhanced FM and standard QFM operation (**e**). The ARW and bias instability are evaluated by the lower $\tau^{-1/2}$ and $\tau^0$ limits, respectively.

with the standard QFM set-up, resulting in an ARW of 0.064° (√h)⁻¹ (0.44° (√h)⁻¹) and a bias instability of 2.7° h⁻¹ (15.1° h⁻¹).

The experimentally demonstrated cusp-singularity-enhanced Coriolis effect in FM operation provides large improvements in SNR and precision. This is because the dominant error source is resonant frequency fluctuations[30], which affect the FM output but are not amplified by the singularities. The influence of these errors is greatly suppressed when the FM signal is converted to the angular velocity using the singularity-enhanced sensitivities.

## Singularity-enabled PM measurements

Furthermore, we find that, when the PhT system operates near $X_{1,2}$, the relative phase $\vartheta$ can be a superior metric for rotation readout than the PhT frequency, enabling a PM gyroscope that achieves strategic-grade SNR on silicon chips.

The relative phase $\vartheta$ of the PhT system as a function of angular velocity $\Omega$ and coupling strength $g$ is theoretically illustrated in Fig. 4a (Supplementary Note 9). Notably, $\vartheta$ exhibits cusp singularities at $X_1$: [$g = (\sqrt{2}-1)\gamma, \Omega = 0$] and $X_2$: [$g = (-\sqrt{2}-1)\gamma, \Omega = 0$], coincident with those in the $\omega_T$ domain (see Fig. 1d, in which the colour gradient in the $\omega_T$ surface represents $\vartheta$). In standard QFM operation ($g = 0$), $\vartheta$ remains constant; whereas near the cusps it changes sharply, enabling highly sensitive PM readout.

To demonstrate the $\vartheta$-based sensing, we extract the $\vartheta$ data from the $\omega_T$ measurements in Fig. 3a. The resulting $\vartheta$-versus-$\Omega$ curves over ±0.15° s⁻¹ are shown in Fig. 4b. No differential process is required because $\vartheta$ is intrinsically stable (Extended Data Fig. 2b). On a logarithmic scale (Fig. 4c), the $\Omega$-induced phase modulations at both $X_1$ and $X_2$ exhibit slopes of 1/3, indicating a cubic-root sensitivity. As with the FM outputs, $X_2$ provides a greater sensitivity than $X_1$.

Moreover, an intriguing topological feature emerges in the $\vartheta$ space: $X_1$ and $X_2$ are separated by a phase singularity PS₂. Crossing PS₂ induces a topological phase transition, with $X_2$ ($X_1$) lying in a non-trivial (trivial) phase. As $\Omega$ is swept from $-\infty$ to $+\infty$ through $X_2$ ($X_1$), $\vartheta$ exhibits a net winding of $-2\pi$ (0), as shown in the insets of Fig. 4a. These trajectories correspond to the red and blue paths in Fig. 1e projected onto the $S_1$–$S_2$ plane. These topological signatures are confirmed by wide-range $\Omega$ sweeps (up to ±1,000° s⁻¹) in Fig. 4c and Extended Data Fig. 2d, in which the phase values are unwrapped to avoid discontinuities. This topological distinction suggests that $X_2$ supports a wider sensing range than $X_1$.

The Allan deviations of the bias $\vartheta$ signals (recorded simultaneously with FM bias signals) are shown in the inset of Fig. 4d. The bias $\vartheta$ stability at $X_1$ is about 2.7 times better than that at $X_2$, owing to the larger $|q_2|$ amplitude at $X_1$ (Extended Data Fig. 4) that reduces the measurement noise in $\theta_2$ and, consequently, in $\vartheta$. We calculate the normalized Allan deviations $\sigma_\Omega$ of the PM operation (Fig. 4d) by converting the bias $\vartheta$ signals into equivalent $\Omega$ inputs (Supplementary Note 10 and Extended Data Fig. 3b). The resulting bias instabilities reach 0.035° h⁻¹ at $X_1$ and 0.056° h⁻¹ at $X_2$, approaching inertial grade. Even more impressive is the SNR performance: the strategic-grade ARWs of 0.00036° (√h)⁻¹ at $X_1$ and 0.00057° (√h)⁻¹ at $X_2$ are comparable with those of the bulky, expensive HRGs for high-end applications[4,6] (Fig. 4e).

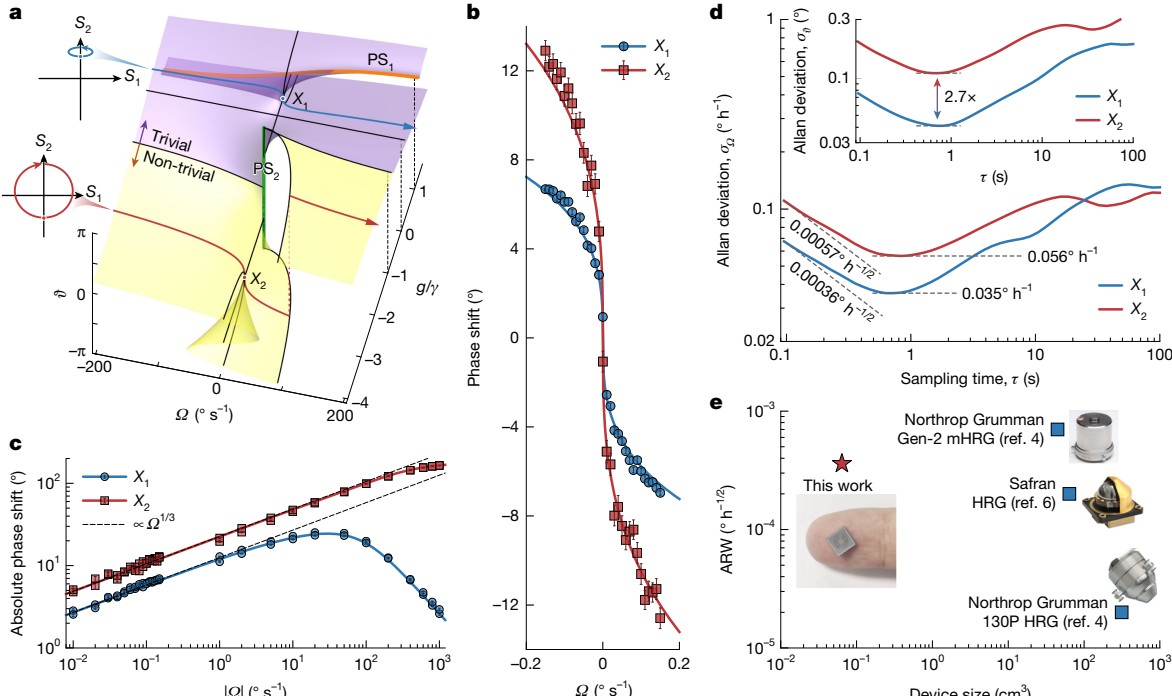

**Fig. 4 | PhT singularity-mediated phase modulation for ultrasensitive rotation measurements. a**, Theoretically computed relative phase $\vartheta$ of the SW modes as a function of angular velocity $\Omega$ and coupling strength $g$, showing two cusp catastrophes. The cusp singularities, $X_1$ and $X_2$, are separated by a phase singularity $PS_2$ and lie within trivial and non-trivial topological phases, respectively. The state vectors projected onto the $S_1$–$S_2$ plane as $\Omega$ changes from negative to positive infinity across $X_1$ or $X_2$ are shown by the insets, depicting distinct topologies. **b**, Observed phase shift of $\vartheta$ (points) as a function of small rotation perturbations when operating near $X_{1,2}$. The phase output data are measured without using the differential configuration. The error bars represent the standard deviations. The red and blue curves are theoretical $\vartheta$ shifts for $g = (-\sqrt{2} - 1)\gamma$ and $(\sqrt{2} - 1)\gamma$, respectively. **c**, Logarithmic plot of the observed (points) and calculated (solid curves) phase shift of $\vartheta$ as a function of angular velocity $\Omega$ for both slow and fast rotation measurements, obtained when the system is initialized at $X_{1,2}$. **d**, Allan deviations of the effective bias inputs for the singularity-mediated zero-$\Omega$ phase outputs. Inset, Allan deviations of the original bias phase outputs. **e**, Comparison of ARW and device size with some typical high-end HRGs[4,6]. Top and bottom right insets adapted with permission from ref. 4, IEEE; middle right inset adapted with permission from ref. 6, IEEE.

This PM operation delivers more than two orders of magnitude improvement in both stability and SNR relative to FM operation. First, unlike $\omega_T$, $\vartheta$ is inherently unaffected by resonant-frequency fluctuations (Supplementary Note 9), thereby enabling higher measurement stability. Second, because frequency noise is integrated into phase noise, the spectral density acquires a $1/\omega_0^2$ suppression factor (Supplementary Note 11), yielding much lower noise in PM readout compared with FM detection.

Crucially, the Brownian-motion-induced phase (frequency) noise enters the PM (FM) outputs additively, without singularity amplification (Supplementary Note 11). By contrast, the singular Coriolis responses follow boosted cubic-root responsivities, enabling SNRs that exceed the fundamental limit of the conventional CVG theory. As a result, the singularity-mediated PM readout surpasses the conventional theoretical performance limits of the resonator, for which the bias instability and ARW are predicted to be >0.32° h$^{-1}$ and >0.018° ($\sqrt{h}$)$^{-1}$, respectively (Supplementary Note 12). These results establish singularity-mediated phase modulation as a fundamentally superior transduction mechanism.

## Discussion and conclusions

For the first time to our knowledge, we have demonstrated how to generate ultrasensitive responses with sublinear scaling of the Coriolis effect by operating near singularities that reside within cusp catastrophes, advancing the previous understanding that the CVG output is always proportional to the rotation input. Through this singular Coriolis effect, we have broken the physical limit of the CVG sensitivity imposed by the intrinsic Coriolis factor. We have shown that this discovery can lead to large improvements in sensitivity, precision and SNR of a CVG. Our findings open up fundamentally new avenues to regulating CVGs and other systems involving the Coriolis effect.

Moreover, by using the PM output of the enhanced Coriolis effect, we have showcased a chip-scale CVG achieving HRG-comparable, strategic-grade ARW, outperforming the current cutting-edge silicon-chip gyroscopes[39–44] by almost an order of magnitude (see Extended Data Table 1 for a detailed comparison). Our findings challenge the traditional view that miniaturized gyroscopes always suffer from reduced SNR, addressing the continuing debate over whether such chip-scale gyroscopes can rival their larger traditional counterparts. Miniaturizing high-performance gyroscopes without losing precision will revolutionize the market, enabling widespread access to advanced navigation and stabilization technologies in affordable, compact devices.

Finally, the substantial enhancements in sensitivity, SNR and precision achieved through the cusp-catastrophe-based singularity enabled by PhT control—among the highest in recent singularity-enhanced sensing experiments[11,16–26] (see Extended Data Table 2 for a detailed comparison)—along with the ultrasensitive singularity-enabled PM operation demonstrated in this study, could lead to advancements in any field requiring extreme sensitivity. The PhT cusp-singularity-enhanced pattern can be adapted to all kinds of sensing application, such as environmental monitoring[45], healthcare sensing[46], seismology[47], gravity measurement[48] and even gravitational-wave detection[49], potentially revolutionizing the development of more sensitive, compact and cost-effective measurement systems.

At present, the demonstrated cusp-singularity-enhanced measurements use a single-channel, non-self-calibrated configuration. Future

implementations that incorporate differential architectures[29,30,50] could further improve bias stability.

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

## Methods

### Experimental realization

The gyroscopic system outlined in Fig. 1a is realized using a 4-mm-diameter on-chip disc resonator exhibiting 12-fold symmetry. Extended Data Fig. 1a shows a top-view photograph, highlighting the resonator in yellow. The deformable structure of the resonator features ten concentric rings linked by radial spokes, with its central part bonded to a substrate. Fabricated from 100-μm-thick P-type single-crystal silicon, the resonator is housed in a vacuum environment to reduce air damping. The resonator and associated circuitry are integrated on a printed circuit board mounted on a temperature-controlled ($25 \pm 0.05$ °C) angular rate table. This set-up allows accurate out-of-plane rotations with precise angular velocities (error $< 0.001°$ s$^{-1}$). Electric connections are maintained during rotation using slip rings, linking the rotating parts with external equipment.

The elastic deformations in the rings and spokes of the silicon disc resonator lead to several eigenstates. This research uses a pair of nearly degenerate in-plane wineglass SW modes with wavenumber $n = 3$ (that is, with $2 \times n = 6$ nodes or antinodes arranged circularly), designated as modes 1 and 2 in Extended Data Fig. 1b. Each SW mode is a combination of two in-plane whispering-gallery TW modes sharing the wavenumber $n = 3$, which travel in opposite directions, denoted as CW and counterclockwise (CCW) in Extended Data Fig. 1c. The relationship between SW and TW modes is detailed later.

The fixed nodes and antinodes of SW modes enable actuation, transduction and tuning using fixed capacitive electrodes, situated with uniform 10-μm gaps from the resonator. Electrodes are colour-coded by function in Extended Data Fig. 1a. Differential actuation is achieved by applying antiphase signals to two opposite antinodal electrodes. Modes 1 and 2 are driven simultaneously, with the drive signal of mode 2 phase-shifted by $+\pi/2$ relative to mode 1. The quadrature excitation of these modes creates a CW TW mode. Antinodal displacements of modes 1 and 2 are detected capacitively in a differential set-up using charge amplifiers.

The transduced displacement signals, $q_{1,2}$, are processed by a lock-in amplifier (Zurich Instruments MFLI) for demodulation against the reference driving signal, yielding the amplitudes $|q_{1,2}|$ and phases $\theta_{1,2}$. The relative phase $\vartheta$ between $\theta_2$ and $\theta_1$ is also recorded. Open-loop measurements use the parametric sweeper block of the lock-in amplifier, sequentially adjusting the reference driving frequency $\omega_d$ while monitoring the amplitudes and phases as functions of $\omega_d$, that is, $|q_{1,2}|(\omega_d)$, $\theta_{1,2}(\omega_d)$. In closed-loop operation, a PLL with a PID controller keeps the phase of mode 1 at $-\pi/2$, by adjusting the oscillation frequency. This PLL-controlled frequency, achieving $\theta_1 = -\pi/2$ phase-tracking, is denoted $\omega_T$.

Imperfections in the degeneracy of the fabricated disc resonator are corrected through a preparatory tuning process using a proven mode-matching technique[51]. This procedure uses two electrode sets, depicted as pink and green in Extended Data Fig. 1a. A DC voltage $V_c$ is implemented on one (for negative $V_c$) or two (for positive $V_c$) off-axis green electrodes between the principal axes of modes 1 and 2 to facilitate stiffness coupling. This results in an electrostatically tunable off-axis spring with stiffness $\Delta_c = E_{1,2}(2V_0 V_c - V_c^2)$, in which $E_1 \approx 10{,}066$ N m$^{-1}$ kg$^{-1}$ V$^{-2}$ or $E_2 \approx 21{,}068$ N m$^{-1}$ kg$^{-1}$ V$^{-2}$ correspond to tuning factors for one or two electrodes[52]. Here $V_0 = 30$ V is the static voltage applied to the resonator body. The off-axis tuning introduces a coupling with strength $g = -\Delta_c/(2\omega_0)$, with the negative sign owing to a roughly $-15°$ azimuthal angle between the tuning electrodes and the principal axis of mode 1. Fabrication flaws may cause slight misalignment of the electrodes with the central axis of modes 1 and 2, resulting in minor non-degeneracy, which is corrected by readjusting the in-axis tuning voltage $V_f$ on the pink electrodes in Extended Data Fig. 1a. The DC voltages $V_0$, $V_c$ and $V_f$ are provided by the precise IT2800 Source Measure Units.

### Sensitivity measurements

In the small-range measurement, the system is carefully tuned at the balanced cusp singularities, $X_1$: ($V_c \approx -0.777$ V, $\Omega \approx 0°$ s$^{-1}$) or $X_2$: ($V_c \approx 2.170$ V, $\Omega \approx 0°$ s$^{-1}$), to enhance gyroscopic sensitivity. The angular velocity, $\Omega$, is varied incrementally from $-0.15°$ s$^{-1}$ to $0.15°$ s$^{-1}$ in $0.01°$ s$^{-1}$ steps, with stabilization at each point for about 120 s to allow the system to settle. The steady-state outputs, PhT frequency $\omega_T$ and relative phase $\vartheta$ are recorded for 5 s, producing 525 data points for each angular velocity, depicted by blue (for $X_1$) or red (for $X_2$) points in Extended Data Fig. 2a,b for $\omega_T$ and $\vartheta$, respectively.

The $\omega_T$ frequency outputs in small-range measurements are strongly affected by resonant-frequency fluctuations. To mitigate these effects, a differential output configuration is used, measuring the difference between each frequency output and a reference output $\omega_T(\Omega_r)$ at a constant angular velocity $\Omega_r = 1.5°$ s$^{-1}$ after each valid output (black points in Extended Data Fig. 2a). These differential values are adjusted by $\omega_T(\Omega_r) - \omega_T(0)$ to shift the origin to zero, with $\omega_T(0)$ being the expected frequency at $\Omega = 0°$ s$^{-1}$. The offset frequency differences $\delta\omega_{X1,2}$ at singularities $X_1$ and $X_2$ are indicated by blue and red points in Fig. 3a, respectively. Each point represents the average of 525 samples, with error bars showing the standard deviation.

By contrast, the phase outputs $\vartheta$ for small-range measurements exhibit much greater stability, as shown in Extended Data Fig. 2b. Thus, no differential operation is applied to the $\vartheta$ outputs, notwithstanding that reference outputs $\vartheta(\Omega_r)$ have been recorded (black points in Extended Data Fig. 2b). The mean and standard deviation of 525 $\vartheta$ samples for each angular velocity input are depicted as points with error bars in Fig. 4b.

For extensive measurements, the angular velocity is sequentially set to $\Omega \in \{0, \mp1, \mp2, \mp5, \mp10, \mp20, \mp50, \mp100, \mp200, \mp400, \mp600, \mp800, \mp1{,}000\}°$ s$^{-1}$, following system calibration to balanced cusp singularities. The stabilized PhT frequency $\omega_T$ and relative phase $\vartheta$ are maintained for 5 s to capture 525 data points at each angular velocity. Mean values for $\omega_T$ and $\vartheta$ across angular velocities are represented by blue ($X_1$) and red ($X_2$) points in Extended Data Fig. 2c,d, with standard deviations as error bars. The $\vartheta$ values are unwrapped to prevent discontinuities. Similarly, measurements are performed with the standard QFM mode at $V_c = 0$, using angular velocities $\Omega \in \{0, \mp200, \mp400, \mp600, \mp800, \mp1{,}000\}°$ s$^{-1}$, recording only $\omega_T$, shown by black points in Extended Data Fig. 2c and Fig. 3b.

### Data availability

All data are available on figshare[53] (https://doi.org/10.6084/m9.figshare.29278061.v3). Source data are provided with this paper.

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

**Acknowledgements** X.Z. thanks A. Seshia from the University of Cambridge for discussions. This work is partly supported by the National Key R&D Program grants 2024YFE0102400 (H.J.) and 2022YFB3204901 (X.Z.), the National Natural Science Foundation of China (NSFC) grants U21A20505 (D.X., X.Z. and X.W.), 11935006 (H.J.), 12421005 (H.J.) and 62174077 (F.W.), the Hunan Major Sci-Tech Program grant 2023ZJ1010 (H.J.), the RIKEN Special Postdoctoral Researchers (SPDR) programme (R.H.), Young Elite Scientist Sponsorship Program by the China Association for Science and Technology (CAST) grant YESS20200127 (X.Z.) and the Natural Science Foundation of Hunan Province for Excellent Young Scientists grant 2021JJ20049 (X.Z.). This work is primarily supported by the National Natural Science Foundation of China (NSFC) grant 52575679 (X.Z.).

**Author contributions** F.N., H.J. and X.Z. initiated the research. X.Z. conceived the idea. X.Z. and S.Z. performed the experiments. X.Z. processed the data. X.Z., H.J. and F.N. conducted the theory. X.Z. designed the device. L.Y., N.Z., K.H. and X.Z. fabricated the device. X.Z., S.Z., F.W., D.X. and X.W. developed the test system. X.Z., H.J. and F.N. wrote the manuscript, with input from all authors. H.J., F.N., R.H., F.W. and X.Z. revised the manuscript. H.J., F.N., F.W. and X.Z. jointly supervised the project.

**Competing interests** The authors declare no competing interests.

**Additional information**
**Correspondence and requests for materials** should be addressed to Fei Wang, Franco Nori,
Hui Jing or Xin Zhou.
**Peer review information** *Nature* thanks Tsampikos Kottos and the other, anonymous,
reviewer(s) for their contribution to the peer review of this work. Peer reviewer reports are
available.

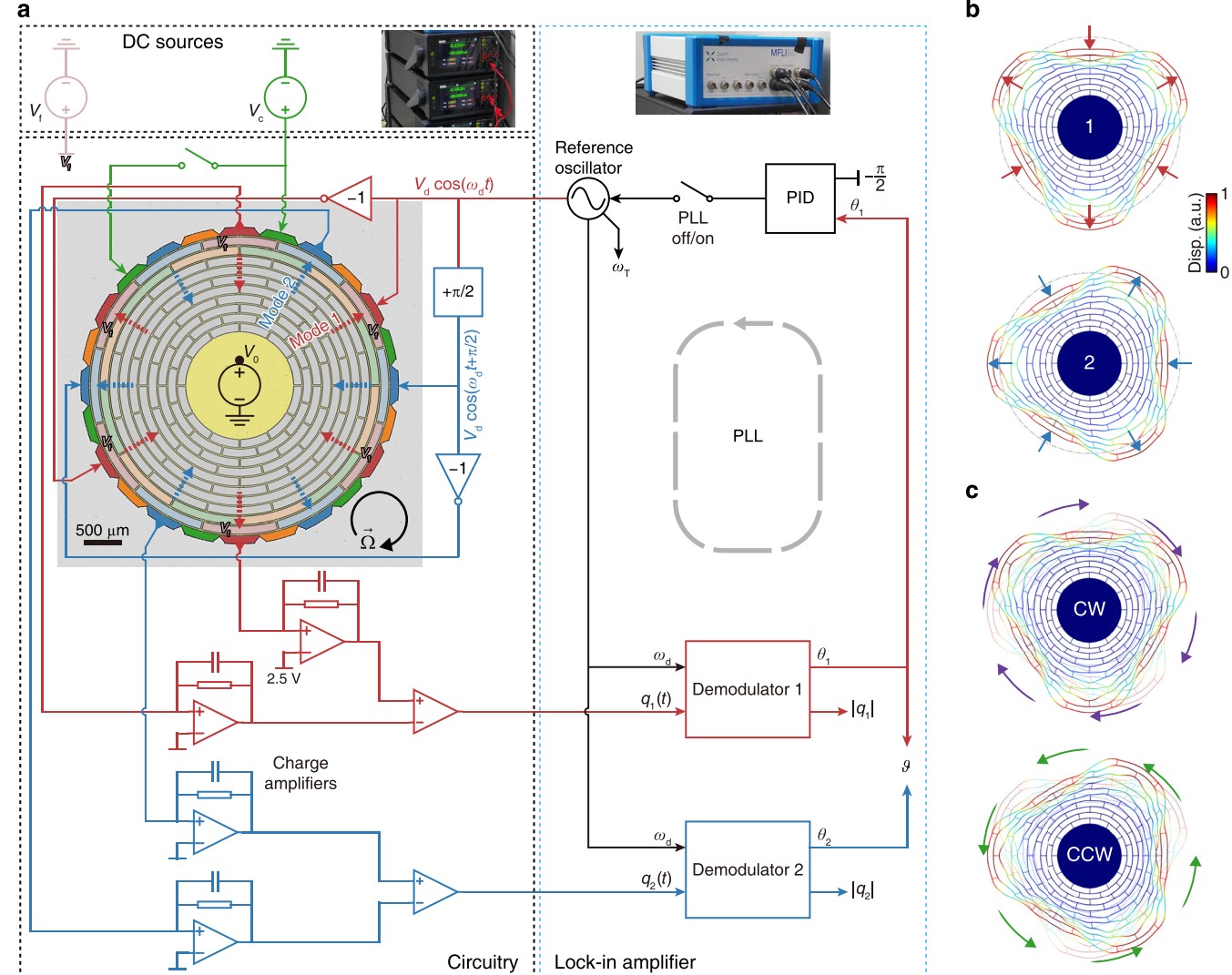

**Extended Data Fig. 1 | Experimental realization. a**, Image and schematic of the experimental configuration. The picture of the disc resonator device is a pseudo-coloured microscope photograph. The device, along with its signal processing circuitry, is embedded on a printed circuit board mounted on an angular rate table. A lock-in amplifier facilitates drive, detection and PLL control. Electrostatic tuning is accomplished through the application of precise DC voltages. **b**, Mode shapes with exaggerated amplitudes for SW modes 1 and 2. **c**, Diagram of CW and CCW TW modes.

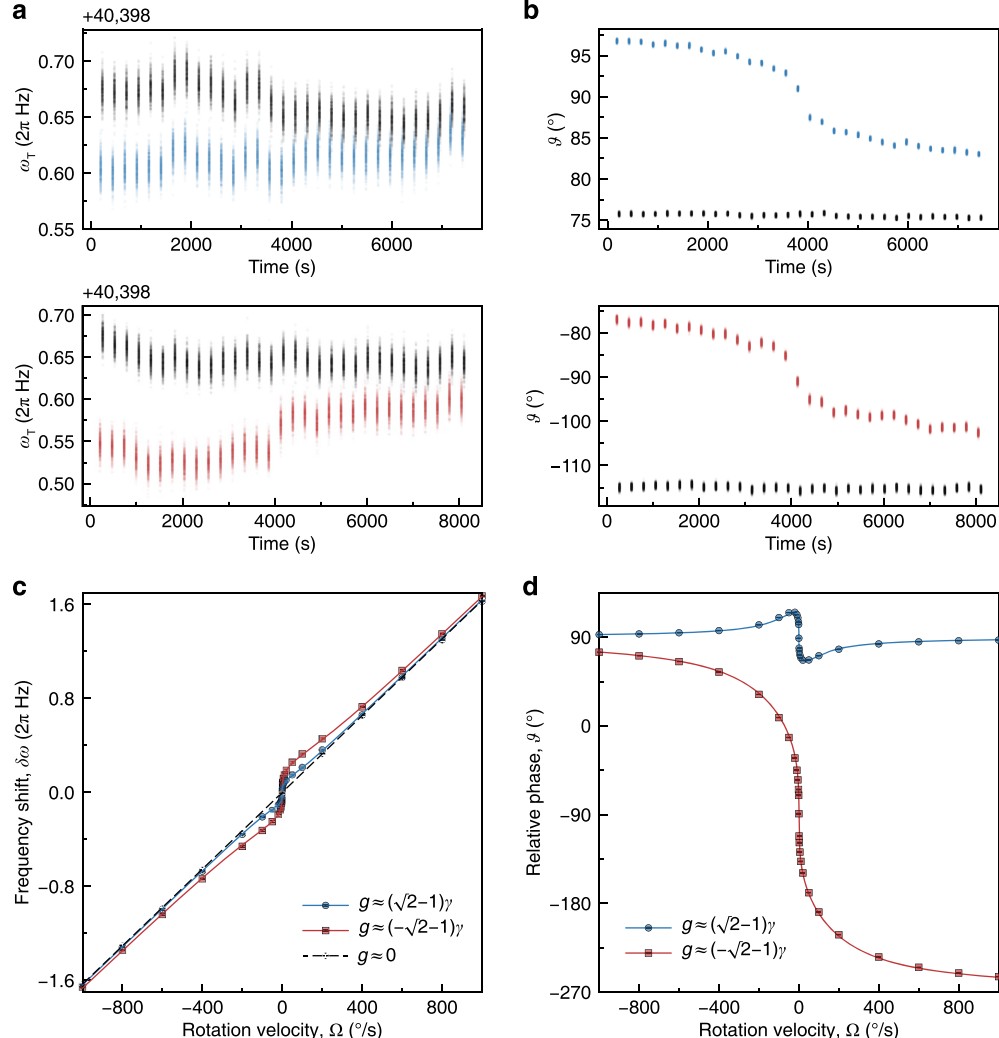

**Extended Data Fig. 2 | Measurements of the PhT frequency and relative phase as dependents on angular velocity. a**, The PhT frequency $\omega_T$ as a function of angular velocity $\Omega \in \{-0.15, -0.14,..., 0,..., 0.14, 0.15\}°\,s^{-1}$ with the system calibrated to balanced cusp singularities $X_1$ (blue) and $X_2$ (red). Each cluster comprises 525 data points of $\omega_T$ at the same $\Omega$ input. Black markers represent $\omega_T(\Omega_r)$ measured at a constant reference angular velocity $\Omega_r = 1.5°\,s^{-1}$. **b**, Relative phase $\vartheta$ corresponding to the measurements in **a**. **c**, Shifts in the PhT frequency $\omega_T$ as a function of angular velocity in large-range measurements when the system is tuned to operate at $X_1[g \approx (\sqrt{2}-1)\gamma]$, $X_2[g \approx (-\sqrt{2}-1)\gamma]$ or standard QFM mode ($g \approx 0$). **d**, Relative phase $\vartheta$ corresponding to the measurements in **c** operated at $X_{1,2}$. The error bars represent the standard deviations.

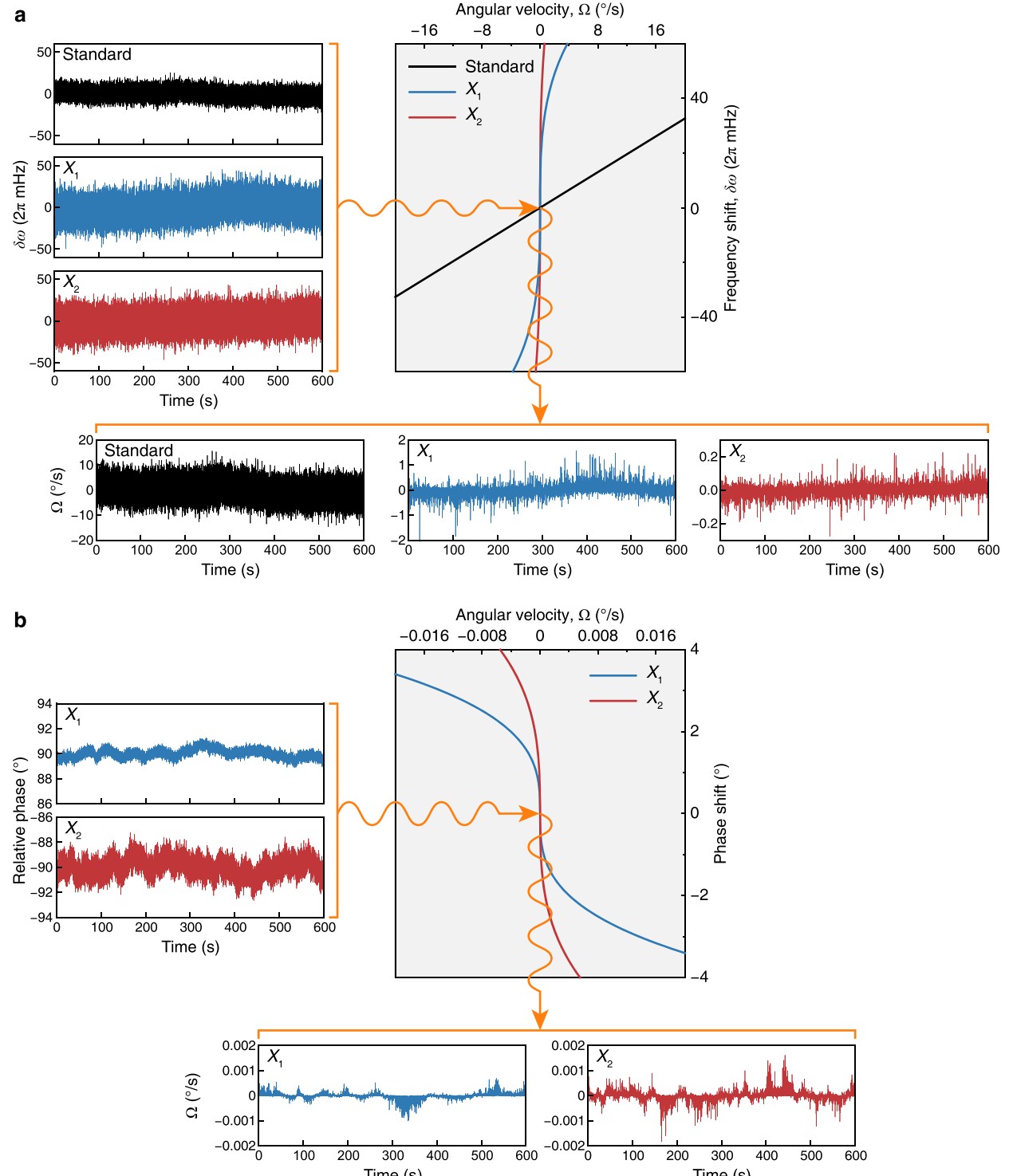

**Extended Data Fig. 3 | Bias outputs and estimated effective angular-velocity inputs. a**, Zero-rotation bias signals of the PhT frequency $\omega_T$ when the system is operated in standard QFM mode and at the balanced cusp singularities $X_{1,2}$. Effective angular-velocity inputs corresponding to the bias output signals are estimated using the $\Omega$ to $\omega_T$ transduction functions. **b**, Bias signals of the relative phase $\vartheta$ corresponding to the $\omega_T$-bias measurements at $X_{1,2}$ in **a**. Effective inputs are estimated using the $\Omega$ to $\vartheta$ transduction functions. The $\Omega$ estimation processes are detailed in Supplementary Note 10.

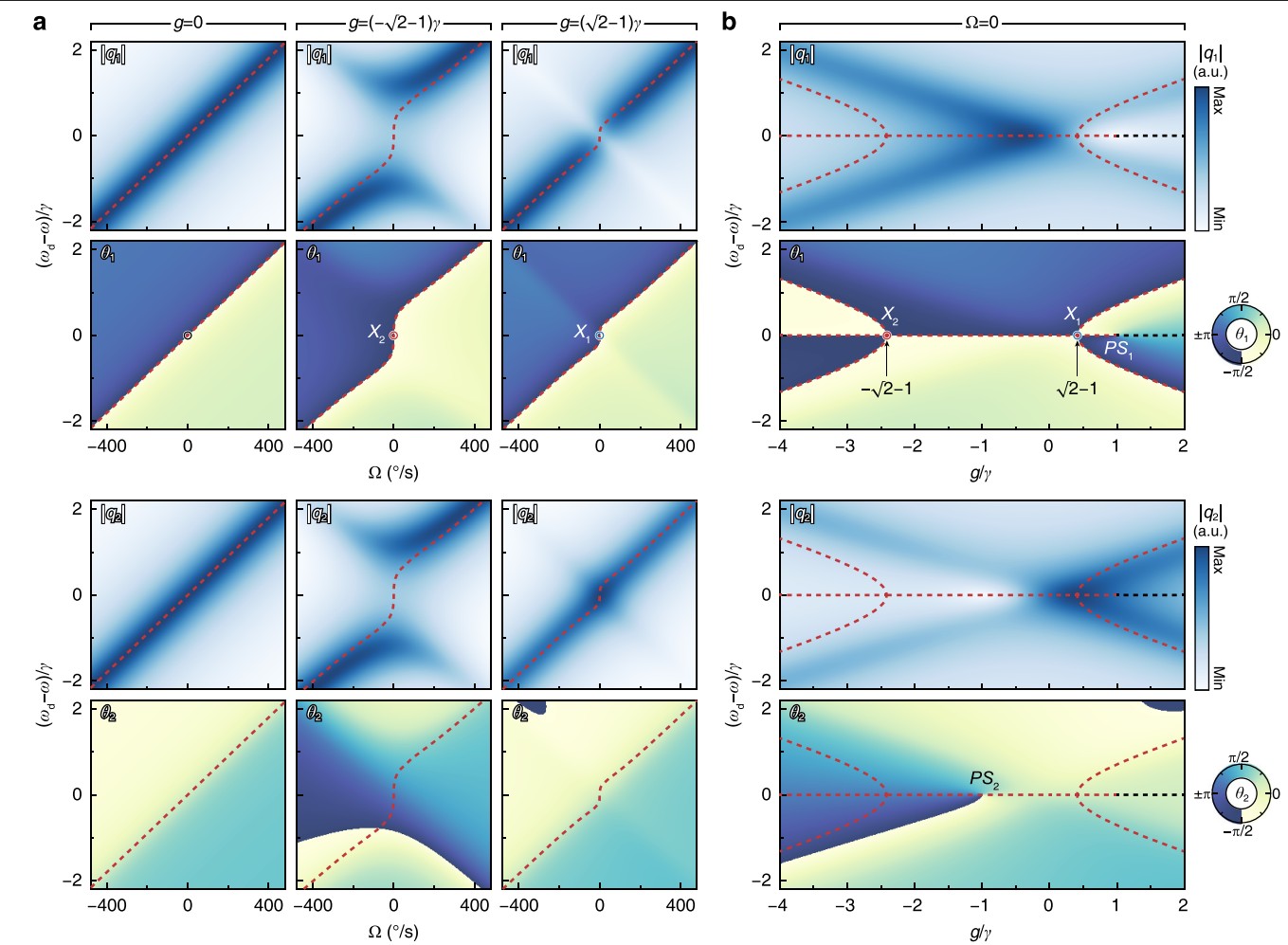

**Extended Data Fig. 4 | Theoretical frequency responses of SW modes, $|q_{1,2}|(\omega_d)$ and $\theta_{1,2}(\omega_d)$. a**, The amplitudes $|q_{1,2}|(\omega_d)$ and phases $\theta_{1,2}(\omega_d)$ are depicted as functions of $\Omega$ under the conditions $g = 0$, $(-\sqrt{2}-1)\gamma$ and $(\sqrt{2}-1)\gamma$.

**b**, The $|q_{1,2}|(\omega_d)$ and $\theta_{1,2}(\omega_d)$ are shown as functions of $g$ at fixed $\Omega = 0$. The red (black) dashed lines denote the equiphase contours at which $\theta_1 = -\pi/2$ ($\pi/2$).

**Extended Data Table 1 | Comparison with state-of-the-art high-performance silicon-chip CVGs**

| Reference | Config. | Structure | Resonant Frequency, Detuning (if any) | Quality factor | Intrinsic Coriolis factor | Proof mass | Drive Ampl. | ARW (º/√h) | Bias stability (º/h) | Year |
|---|---|---|---|---|---|---|---|---|---|---|
| Boeing, JPL [39] | AM | Disk | 14 kHz | 80k | 0.8 | 1.1 mg | 1.8 μm | 0.0021 | 0.012 (Allan) | 2014 |
| University of California, Irvine [40] | AM | Quad-mass tuning fork | 1.6 kHz | 1.8M | 0.87 | 8 mg | 2.5 μm | 0.015 | 0.09 (Allan) | 2016 |
| Honeywell [41] | AM | Dual-mass tuning fork | --, >700 Hz | -- | >0.9 | -- | -- | 0.003 | 0.009 (Allan) 0.05 (1σ@10s) | 2019 |
| Northrop Grumman LITEF GmbH [42] | AM | Dual-mass tuning fork | -- | -- | >0.9 | -- | -- | 0.02 | 0.007 (Allan) 0.3 (1σ@10s) | 2019 |
| Politecnico di Milano [43] | AM | Dual-mass tuning fork | 25 kHz, 100-200 Hz | 25k-50k | >0.9 | 3.5 μg | 9 μm | 0.004 | 0.02 (Allan) | 2021 |
| Silicon Sensing CRS39A | AM | Ring | 14 kHz | -- | 0.8 | -- | -- | 0.004 | 0.03 (Allan) | 2021 |
| NUDT [44] | AM | Disk | 4.2k | 570k | 0.85 | 1.8 mg | 2 μm | 0.003 | 0.015 (Allan, standard), 0.003 (Allan, mode reversal&deflection) 0.08 (1σ@10s, mode reversal&deflection) | 2025 |
| This work | Singularity enhanced | Disk | 40.4kHz | 112k | 0.588 | 60 μg | 0.5 μm | 0.00036 | 0.035 (Allan) | 2025 |

Data from refs. 39–44.

**Extended Data Table 2 | Comparison with previous experiments of singularity-enhanced sensing**

| Reference | Singularity type (order) | Degrees of Freedom | Enhancement object | Enhancements | | | Year |
|---|---|---|---|---|---|---|---|
| | | | | Responsivity | SNR | Precision | |
| Chen et.al. [16] | Exceptional point (2) | 2 | Nanoparticle sensing | 2.5× | -- | -- | 2017 |
| Hodaei et.al. [17] | Exceptional point (3) | 3 | Joule heat sensing | 23× | -- | -- | 2017 |
| Hokmabadi et.al. [20] | Exceptional point (2) | 2 | Sagnac effect | 20× | -- | -- | 2019 |
| Lai et.al. [21] | Exceptional point (2) | 2 | Sagnac effect | 4× | Ineffective | Ineffective | 2019 |
| Kononchuk et.al. [11] | Wigner's cusp anomaly (2) | 2 | Acceleration sensing | 60× | 8.4× | 7.8× | 2021 |
| Kononchuk et.al. [23] | Exceptional point (2) | 2 | Acceleration sensing | 10× | 3× | ~3× | 2022 |
| Suntharalingam et.al. [24] | Nonlinear exceptional point (2) | 2 | Voltage sensing | 100× | >150× | >4.5× | 2023 |
| Xu et.al. [25] | Exceptional point (2) | 2 | Nanometrology | 86× | 5× | ~8× | 2024 |
| Ruan et.al. [26] | Exceptional point (2) | 2 | Magneto-optical effect | 10× | 3× | -- | 2024 |
| This work | Cusp singularity (3) | 2 | Coriolis effect | 1010× | 253× | 297× | 2025 |

Data from refs. 11,16,17,20,21,23–26.