## [Peer Review file · Nature]

Cusp-singularity-enhanced Coriolis effect for sensitive chip-scale gyroscopes

Corresponding Author: Professor Hui Jing

Version 0:

Reviewer comments:

Referee #1

(Remarks to the Author)

This paper proposes and experimentally demonstrates the utilization of a third-order singularity in a Coriolis vibratory gyroscopes (CVG) as a means to enhance the sensitivity and the signal-to-noise ratio (SNR) of silicon-based chip-scale gyroscopes. The paper is well written and the results are scientifically sound. The analysis of noise is standard using the Allan variance and the measurements are very convincing. Taken the fact that high-precision inertia sensors are of extreme interest for a variety of applications, the achievement of a world-record SNR performance of such chip-scale gyroscope constitutes an important result. Of course, such 3-rd order singularity has a caveat i.e. the sensor is not self-calibrated (the user has to identify the perturbation strength with respect to the origin i.e. the singular point). This is in contrast to, say, exceptional points of order 2 (EP-2) where sensing is self-calibrated i.e. the perturbation strength is measured with respect to $\Delta\omega = \omega_{+} - \omega_{-} \sim \sqrt{\epsilon}$, where ω_{\pm} are the detuned eigenfrequencies from the EPD and ϵ is the small perturbations. The authors has to point out this "deficiency" of their proposed scheme either in the conclusions or in the introduction.

Some other comments: (a) it might be beneficial for the reader to have a short derivation of the third-order singularity in the main text (not mandatory comment and only assuming that space is allowed); (b) The authors might consider to include the reviews [Nature Communications volume 11, Article number: 2454 (2020)] and [Photon. Res. 8, 1457-1467 (2020)]. (c) They might also want to cite also the recent paper [Phys. Rev. Applied 23, 064043].

Once these comments are taken into consideration and addressed, I will be happy to recommend the paper for publication to Nature

Referee #2

(Remarks to the Author)

Authors claim to use third-order singularities lying within cusp catastrophes in the phase-tracked oscillations of an on-chip CVG to facilitate a cubic-root scaling of the Coriolis-effect-induced frequency modulation. The idea appears novel. I have however the following comments.

1
"However, the reduced sensitivity of miniaturized sensing elements constrains the performance of chip-scale CVGs, confining them to low-end applications"

What do authors mean with reduced sensitivity? Microscale gyros can have scale factors identical to macroscale (e.g. lumped mass RIG have even larger angular gain than HRGs) and even lower noise (e.g. sub 100-udps/rt-Hz is achieved). The critical point for microgyro is instead related to their imperfections and consequent asymmetries at the microscale, inducing e.g. cross-stiffness, cross-damping, eccc...

2

"as the inherently weak Coriolis force usually necessitates large-scale structures to produce noticeable effects"

Again, the angular gain is the same and the force effect is not weaker.

Additionally, transduction of micro-scale CVG is way larger due to miniaturized gaps.

Again, the benefit of macroscale HRGs is their material (quartz), long term stability (quartz, again) and good matching between modes, often trimmed via laser effects or so. This is not convenient for large volume production (cost etc). But it is not the weak Coriolis force the major issue.

Please rephrase all the initial portion of the paper. It can be still useful to enhance the scale factor, but the mentioned reasons are wrong.

3

"This coefficient measures the proportion of modal mass contributing to the Coriolis effect and is intrinsically determined by 25 vibratory-mode geometry, always constrained to ≤ 1 ".

As a further confirmation of the points above, this is valid also for macroscale devices!

4

The baseline for the authors study, which is a QFM, is known to be not good for long term navigation purposes due to the drifts of the frequencies.

Besides, note that in a QFM both the frequencies change as $\omega_0 + k_0 \cdot \Omega$.

5

The introduced idea is new but... A starting situation with a bias stability of 800 °/hr is not at all the state of the art (even in my pocket I have better gyros in the mobile phone). There are QFM and LFM reaching bias stability of 1 °/hr or sub °/hr.

Additionally, there are AM gyroscopes reaching 0.00x °/hr.

So, the 2xx-fold improvement shown by the authors is true only against a very poor starting sensor.

As such, in my opinion the paper cannot be accepted, because final results lie well below state of the art performance of microgyroscopes.

Version 1:

Reviewer comments:

Referee #2

(Remarks to the Author)

Thank you for answering the different points, and for providing extensive explanations.

The work can now be accepted.

As a collateral comment: please take criticism from reviewers positively, as a way to improve the manuscript. This is exactly what happened. Reviews may come from people who are very very expert in the field and you need sometimes to humbly accept their comments.

Response to Referees

We express our sincere thanks to the reviewers for their thoughtful and constructive feedback, which has greatly improved the quality of the manuscript. Below, we address each comment point by point. The manuscript has been revised to incorporate all the comments. The original title “Enhancement of the Coriolis effect via cusp singularities for ultrasensitive chip-scale gyroscopes” has been revised as “**Cusp-singularity-enhanced Coriolis effect for ultrasensitive chip-scale gyroscopes**”. For clarity, reviewers’ comments appear in **blue**, authors’ responses in **black**, and the corresponding changes to the manuscript are highlighted in **red**.

Referees' comments:

Referee #1 (Remarks to the Author):

Comment 1.0: This paper proposes and experimentally demonstrates the utilization of a third-order singularity in a Coriolis vibratory gyroscopes (CVG) as a means to enhance the sensitivity and the signal-to-noise ratio (SNR) of silicon-based chip-scale gyroscopes. The paper is well written and the results are scientifically sound. The analysis of noise is standard using the Allan variance and the measurements are very convincing. Taken the fact that high-precision inertia sensors are of extreme interest for a variety of applications, the achievement of a world-record SNR performance of such chip-scale gyroscope constitutes an important result. Of course, such 3-rd order singularity has a caveat i.e. the sensor is not self-calibrated (the user has to identify the perturbation strengt with respect to the origin i.e. the singular point). This is in contrast to, say, exceptional points of order 2 (EP-2) where sensing is self-calibrated i.e. the perturbation strength is measured with respect to $\Delta\omega = \omega_{+} - \omega_{-} \sim \sqrt{\varepsilon}$, where ω_{\pm} are the detuned eigenfrequencies from the EPD and ε is the small pertunations. The authors has to point out this "deficiency" of their proposed scheme either in the conclusions or in the inrtoduction.

Response: We sincerely thank the Referee for the positive and encouraging assessment. We are grateful for your recognition that the work is scientifically sound, convincing, and important, and for highlighting the significance of achieving record SNR in a silicon chip CVG for precise inertial sensing.

We also appreciate your thoughtful observation that, unlike second-order exceptional points, the third-order cusp singularity is not intrinsically self-calibrating. This is indeed a widely existing problem in the gyroscope field.

To clarify, a large number of practical gyroscopes operate in a locked single-channel configuration. Gyroscopes typically do not measure spectra and compute frequency splits between resonances. Instead, Coriolis vibratory gyroscopes typically lock into resonance using phase-locked loops (PLLs), as we have done in our study, to provide **autonomous** and **continuous** readout. However, a control system can only lock into a single resonance at a time, so many widely used gyroscopes operate in a single-channel locked state.

If higher stability is required, differential techniques can be introduced. For example, a nanophotonic optical gyroscope [Khial, P.P., White, A.D. Hajimiri, A. Nanophotonic optical gyroscope with reciprocal sensitivity enhancement, *Nature Photon* 12, 671–675 (2018)] and a quadrature frequency-modulated CVG [B. Eminoglu, et al., Comparison of long-term stability of AM versus FM gyroscopes. *MEMS* (2016) pp. 954–957] have been operated alternately in opposite (clockwise/counterclockwise) modes to obtain differential outputs with reduced drift.

Fig. R1. By alternately operating a nanophotonic optical gyroscope in clockwise (CW) and counterclockwise (CCW) modes, improved stability is obtained by differentiating the opposite outputs.

In another instance, Kline *et al.* simultaneously implemented a nearby dual-channel CW and CCW quadrature FM operations on one chip to realize a differential readout, improving bias stability by approximately fourfold [M. Kline, et. al. Quadrature FM gyroscope, MEMS (2013) pp. 604–608].

Fig. R2. By simultaneously constructing two opposite quadrature frequency-modulated operations in a dual-resonator chip, stability is improved by about four times.

These differential techniques can be directly applicable to our cusp-singularity-enhanced operation and could enable self-calibrated outputs with additional performance gains. However, we view such engineering refinements as promising directions for follow-up studies. The present work focuses on establishing the new operating principle, cusp-singularity-enhanced Coriolis effect, and demonstrating its potential to surpass conventional SNR limits.

Changes made:

As the Referee suggested, we have made an additional discussion about the self-calibration issue of the singularity-enhanced operations in the revised Section “Discussion and conclusions”:

“At present, the demonstrated cusp-singularity-enhanced measurements employ a single-channel, non-self-calibrated configuration. Future implementations that incorporate differential architectures^{29,30,50} could further improve bias stability.”

Comment 1.1: Some other comments, (a) it might be beneficial for the reader to have a short derivation of the third-order singularity in the main text (not

mandatory comment and only assuming that space is allowed); (b) The authors might consider to include the reviews [Nature Communications volume 11, Article number: 2454 (2020)] and [Photon. Res. 8, 1457-1467 (2020)]. (c) They might also want to cite also the recent paper [Phys. Rev. Applied 23, 064043].

Response: Thank you very much for the constructive suggestions. As recommended, we have added a concise derivation of the third-order singularity to the main text. We also appreciate your kind help in recommending these important references, which have been added to the revised manuscript.

Changes made:

(a) On page 6, in the Section “**Concept**” of the revised manuscript, we have added a concise derivation of the third-order singularity:

“The coupled dynamics are fully described by the complex susceptibilities of modes 1 and 2 $\chi_{1,2}$. Here, the mode-1 phase $\theta_1 \equiv \text{Arg}(\chi_1)$ is tracked to $-\pi/2$, the corresponding PhT frequency as a function of the stiffness coupling g and angular velocity Ω is determined by the condition $\text{Re}(\chi_1) = 0$ and obeys the cubic equation (Supplementary Note 3),

$$\delta\omega^3 + \kappa_0\Omega\delta\omega^2 + \left(\frac{\gamma^2}{4} - \frac{g\gamma}{2} - \frac{g^2}{4} - \kappa_0^2\Omega^2\right)\delta\omega - \kappa_0\Omega\left(\frac{\gamma^2}{4} + \frac{g^2}{4} + \kappa_0^2\Omega^2\right) = 0, \quad (2)$$

where $\delta\omega \equiv \omega_T - \omega$ is the frequency modulation. Plotting ω_T over the (g, Ω) space under the constraint $\text{Im}(\chi_1) < 0$ yields a partially folded surface exhibiting two cusp catastrophes^{13,31-37}, which describe hysteretic behaviors controlled by two factors, as show in Fig. 1d. The portion with $\text{Im}(\chi_1) > 0$ (the middle sheet for $g > \gamma$) is omitted as it corresponds to the $\theta_1 = \pi/2$ contour and is unattainable for the $\theta_1 = -\pi/2$ PhT control. The boundary defined by $\text{Im}(\chi_1) = 0$ signifies a θ_1 -phase singularity³⁸, PS_1 , indicated by the orange rod at $g = \gamma$.”

(b) The recommended references are included as ref. [18,19,27] of the revised manuscript:

18. J. Wiersig, Prospects and fundamental limits in exceptional point-based sensing. Nature Communications 11, 2454 (2020).

19. J. Wiersig, Review of exceptional point-based sensors. Photonics Research 8, 1457–1467 (2020).

27. A. Suntharalingam, L. Fernández-Alcázar, P. F. Wagner-Boián, M. Reisner, U.

Kuhl, T. Kottos, Symmetry-violation-driven hysteresis loops as measurands for noise-resilient sensors. *Physical Review Applied* 23, 064043 (2025).

Comment 1.2: Once these comments are taken into consideration and addressed, I will be happy to recommend the paper for publication to Nature

Response: Once again, thank you for your thoughtful and constructive comments. Your feedback has been invaluable in improving the quality of this paper. We have carefully addressed each point and made substantial revisions accordingly, and we hope these changes meet with your approval.

Referee #2 (Remarks to the Author):

Comment 2.0: Authors claim to use third-order singularities lying within cusp catastrophes in the phase-tracked oscillations of an on-chip CVG to facilitate a cubic-root scaling of the Coriolis-effect-induced frequency modulation. The idea appears novel. I have however the following comments.

Response: Thank you for the thoughtful assessment and for noting the novelty of our study. We appreciate your careful review and constructive feedback. Below, we respond to each of your points in turn and indicate the corresponding revisions made to the manuscript. We hope these clarifications and revisions can address your valuable concerns.

Comment 2.1: "However, the reduced sensitivity of miniaturized sensing elements constrains the performance of chip-scale CVGs, confining them to low-end applications." What do authors mean with reduced sensitivity? Microscale gyros can have scale factors identical to macroscale (e.g. lumped mass RIG have even larger angular gain than HRGs) and even lower noise (e.g. sub 100-udps/rt-Hz is achieved). The critical point for microgyro is instead related to their imperfections and consequent asymmetries at the microscale, inducing e.g. cross-stiffness, cross-damping, etc...

Response: We would like to express our sincere thanks to the Referee for the valuable feedback, which is very constructive for revising the manuscript.

2.1.1 On the use of the “sensitivity”

We apologize for the ambiguous representation of the “sensitivity”. In the singularity-enhanced sensing literature, *sensitivity* commonly denotes “the weakest signal measurable at a given detection bandwidth,” i.e., it is tied to the scaling of the signal relative to noise (SNR) (see, e.g., [Wang, H. et al. Petermann-factor sensitivity limit near an exceptional point in a Brillouin ring laser gyroscope. Nat Commun **11**, 1610 (2020)], which directly made the above definition.) We apologize for the confusion this may cause, as in some other circumstances, such as the gyroscope industry, sensitivity often refers to the scale factor instead. We intended to assert that, in practice, the SNR of

miniaturized sensing elements in silicon chip-scale CVGs is lower than that of macroscale HRGs, a point we elaborate further in Response 2.1.4. To avoid misunderstanding, we have revised the introduction accordingly; the specific changes are listed in the “**Changes made**” section.

2.1.2 On angular gain factors in micro- and macroscale gyros

We fully agree that the microscale gyros have angular gain factors (intrinsic Coriolis factors) identical to the macroscale ones. This study does not claim that microscale gyros intrinsically have smaller scale factors. We also acknowledge that while certain microscale designs can provide larger intrinsic Coriolis factors than HRGs, these factors remain ≤ 1 for both classes.

2.1.3 On state-of-the-art noise performance of silicon-chip gyroscopes

Thank you for noting that some microscale silicon-chip gyroscopes have reached sub 100-udps/rt-Hz, (specifically, $\leq 10^{-4}\text{°/s}/\sqrt{\text{Hz}}$) noise floor (ARW). For clarity, the $10^{-4}\text{°/s}/\sqrt{\text{Hz}}$ noise floor corresponds to an ARW of $0.006\text{°}/\sqrt{\text{h}}$ via

$$1\text{°/s}/\sqrt{\text{Hz}} = 1\text{°}/\sqrt{\text{s}} = 60\text{°}/\sqrt{\text{h}}$$

The sub $0.006\text{°}/\sqrt{\text{h}}$ (100-udps/rt-Hz) ARW is in line with the data of the well-known state-of-the-art silicon-chip microscale gyroscopes summarized in Extended Data Table 1. For comparison, representative HRGs report ARWs in the range $0.0007\text{°}/\sqrt{\text{h}}$ to $0.00002\text{°}/\sqrt{\text{h}}$. Obviously, the $0.006\text{°}/\sqrt{\text{h}}$ (100-udps/rt-Hz) ARW, and all the state-of-the-art ARW data of the silicon-chip microscale gyroscopes are orders of magnitude worse than the ARW of macroscale HRGs.

Fig. 4d. Allan deviations of the effective bias inputs for the singularity-mediated zero- Ω phase outputs, showing ARW values of $0.00036\text{°}/\sqrt{\text{h}}$ at X_1 and $0.00057\text{°}/\sqrt{\text{h}}$ at X_2 . Inset: Allan deviations of the original bias phase outputs.

In this study, singularity-mediated phase-modulated operation yields an ARW of $0.00036^\circ/\sqrt{\text{h}}$ (see Fig. 4d of the paper), which is about an order of magnitude better than the sub- $0.006^\circ/\sqrt{\text{h}}$ (100-udps/rt-Hz) benchmark for the silicon-chip microscale gyroscopes. To the best of our knowledge, the ARW of $0.00036^\circ/\sqrt{\text{h}}$ is a world record for silicon-chip gyroscopes, which is the first time that a silicon-chip gyroscope can reach HRG-comparable noise performance.

2.1.4 On the fundamental limit of microgyroscope performance

We agree that, at the current stage, the precision of state-of-the-art microscale gyroscopes is closely related to the fabrication-induced imperfections and asymmetries (e.g., cross-stiffness, cross-damping, mode frequency mismatch). Notably, the PhT cusp singularity is also very helpful in mitigating certain imperfections: we have demonstrated that we are able to realize mode matching precision exceeding 1/160 of the mode bandwidth γ [Zhou, X. et al. Higher-order singularities in phase-tracked electromechanical oscillators. *Nat Commun* 14, 7944 (2023). Zhang S. et al. Singularity-enhanced deep sub-linewidth mode matching of the disk resonator gyroscope, *IEEE SENSORS 2024*, 1-4, (2024)]. We believe this cusp singularity has the potential to regulate more kinds of structural imperfections. However, this is a totally different story from this work.

Back to this study, based on the conventional gyroscope theory, the SNR (characterized by the ARW) sets a more fundamental limit for microscale gyroscopes to reach high-end performance. Please allow us to elaborate upon this argument. The SNR is determined by the weakest rotation that can be identified from the noise floor of the gyroscope at a given detection bandwidth. The most fundamental noise in the CVGs comes from the Brownian motion of the sensing elements, which sets the ultimate limit for the gyroscope SNR. Following the standard formulation, the Brownian noise floor of a CVG is given by [F. Ayazi and K. Najafi, *JMEMS* 10, 169 (2001). M. Kline, *Frequency Modulated Gyroscopes*, Ph.D. thesis, University of California, Berkeley, (2015).]

$$\text{ARW}_{\text{Brownian}} \propto \frac{1}{\kappa_0 |q_{\text{dirve}}|} \sqrt{\frac{k_B T}{\omega_0 m Q}} \quad (\text{R1})$$

where κ_0 is the intrinsic Coriolis factor, $|q_{\text{dirve}}|$ is the driven-mode displacement amplitude, k_B and T are Boltzmann's constant and the ambient temperature. The

resonant frequency, effective proof mass, and quality factor are denoted by ω_0 , m , and Q , respectively.

To better see how device size affects $\text{ARW}_{\text{Brownian}}$, we reshape equation (R1) by using the following relationships: $\gamma = \omega_0/Q$ and $k = \omega_0^2 m$, with γ the dissipation rate and k the absolute stiffness of the operational mode,

$$\text{ARW}_{\text{Brownian}} \propto \frac{1}{\kappa_0 |q_{\text{drive}}|} \sqrt{\frac{k_B T \gamma}{k}} \quad (\text{R2})$$

Compared to the macroscale gyroscopes, the microscale ones exhibit worse $\text{ARW}_{\text{Brownian}}$ because it provides a lower drive amplitude $|q_{\text{drive}}|$, and orders-of-magnitude lower absolute stiffness k . This distinction inherently originates from the reduction in size of the sensing element. Moreover, the silicon-chip microscale gyroscopes show much higher dissipation γ (or much lower decaying-time constant $\tau = 2/\gamma$) than macroscale HRGs, though the reason is complicated, which may be related to materials, geometry, size, and so on. The practically displayed higher dissipation further degrades the $\text{ARW}_{\text{Brownian}}$ in microscales.

In summary, based on the traditional gyroscope theory, the silicon-chip microgyroscopes have a lower intrinsic performance limit than macroscale HRGs even regardless of their worse structural imperfections, due to the inherent Brownian-noise increase, and therefore reduced SNR, induced by sensing-element miniaturization.

Changes made:

In light of the valuable feedback, we have revised the confusing statement “However, the reduced sensitivity of miniaturized sensing elements constrains the performance of chip-scale CVGs, confining them to low-end applications.” to avoid conflating scale factor with SNR and to reflect the practical limitations more accurately. The updated paragraph in the Introduction now states:

“Lately, chip-scale CVGs have been developed for much broader uses, including movement monitoring and stabilization control of consumer electronics, automobiles, and more^{1,2}, due to their reduced size, weight, and cost compared to traditional CVGs. Despite these advantages, chip-scale CVGs still lag behind HRGs in performance, making them suitable only for medium- or low-end applications. Enhancing chip-scale CVG performance to match the level of the current HRGs, while preserving their miniaturization and affordability, is highly desirable to enable revolutionary technologies such as GPS-denied personal navigation, XXXXXXXXXX, and

microsatellites. Yet, this goal remains elusive due to significant challenges in microfabrication errors and, more fundamentally, the Brownian noise that increases as sensor dimensions shrink, which in turn degrades the signal-to-noise ratio (SNR).”

Comment 2.2: "as the inherently weak Coriolis force usually necessitates large-scale structures to produce noticeable effects"

Again, the angular gain is the same and the force effect is not weaker.

Additionally, transduction of micro-scale CVG is way larger due to miniaturized gaps.

Again, the benefit of macroscale HRGs is their material (quartz), long term stability (quartz, again) and good matching between modes, often trimmed via laser effects or so. This is not convenient for large volume production (cost etc). But it is not the weak Coriolis force the major issue.

Please rephrase all the initial portion of the paper. It can be still useful to enhance the scale factor, but the mentioned reasons are wrong.

Response: Thank you for your thoughtful comments, and please accept our apologies for the unclear wording in the original manuscript.

First, to avoid any misunderstanding: it is common sense in the field that the angular gain (intrinsic Coriolis factor) is the same for micro- and macro-scale gyroscopes and is only determined by the resonator geometry. In our original wording “as the inherently weak Coriolis force usually necessitates large-scale structures to produce noticeable effects”, we intended to convey that the Coriolis effect is weak for small rotation rates and becomes most apparent in large-scale geophysical flows (e.g., atmospheric or oceanic motion). We did **not** mean that the Coriolis effect is intrinsically weaker at the microscale than at the macroscale. We recognize that our original phrasing could be misleading and have revised the text to avoid confusion (see **Changes made**).

What we wish to emphasize is that the efficiency of the Coriolis interaction, captured by the intrinsic Coriolis factor (angular gain), is fundamentally limited to $\kappa_0 \leq 1$. Consequently, for small rotation rates Ω , the Coriolis coupling strength $2\kappa_0\Omega$ is very weak. In microscale silicon-chip gyroscopes, thermomechanical (Brownian) noise is more pronounced than in macroscale HRGs, so the small physical modulations

(e.g., amplitude or frequency shifts) induced by the Coriolis effect can be easily blurred by the higher Brownian noise floor of chip-scale CVGs compared with macroscale HRGs. This signal-to-noise limitation confines typical chip-scale CVGs to medium- or low-end performance.

We agree that microscale silicon chip gyroscopes can realize narrower capacitive gaps than macroscale devices, which can improve transduction efficiency. However, this potential gain does not compensate for the more severe thermomechanical (Brownian) noise associated with miniaturized sensing elements in silicon chips, as detailed in Response 2.1.4.

In the current state of the art, achieving strong CVG performance generally requires exploiting a high quality factor Q in the operational mode to amplify the weak Coriolis-induced signal. As a result, CVG performance tends to hinge on the resonator's Q . Reaching very high Q at the microscale is challenging and costly, often demanding sophisticated designs, less-integrable non-silicon fabrication processes, and expensive high-vacuum encapsulation. Although notable progress has been made with very-high- Q fused-silica micro-resonators, the Q of state-of-the-art silicon chip gyroscopes remains below what is needed for sub-navigation-grade SNR. In this direction, members of our team have pursued significant Q -enhancement efforts in silicon-chip gyroscopes [X. Zhou et al., Appl. Phys. Lett. 109, 263501 (2016); X. Zhou et al., Phys. Rev. Applied 8, 064033 (2017); Q. Li et al., Microsystems & Nanoengineering 4, 32 (2018)], but these did not yield an SNR as high as in this work.

If one could *directly enhance the Coriolis interaction* itself, surpassing the conventional limit $\kappa_0 \leq 1$, *the resulting Coriolis modulations would be stronger and far less susceptible to being blurred by noise, even in microscale devices with higher Brownian noise*. Our work has made this dream come true.

Specifically, we amplify the Coriolis interaction by operating near third-order singularities in cusp catastrophes, thereby alleviating the need for exceptionally high Q . Phasor analysis (Supplementary Note 11) further shows that the phase or frequency fluctuations from thermomechanical noise add directly to the corresponding output signals. The singularity-enhanced, sublinear Coriolis phase modulation breaks the conventional CVG SNR limit. Using this innovation, we achieve a strategic grade ARW of $0.00036^\circ/\sqrt{\text{h}}$, plus a bias instability of $0.035^\circ/\text{h}$. The SNR performance is approximately an order of magnitude better than state-of-the-art silicon chip gyroscopes.

Changes made:

We appreciate the Referee's careful reading and apologize for our earlier lack of clarity. As suggested, we have revised the **Introduction**, the **Abstract**, the **Supplementary Information**, and **part of the main text** to present our claims more precisely.

(1) Revision to the **Introduction** (third paragraph).

The original phrasing, "as the inherently weak Coriolis force usually necessitates large-scale structures to produce noticeable effects," has been revised as:

"At the heart of this challenge is the limited efficiency of the Coriolis interaction, quantified by the Coriolis factor κ_0 that measures the proportion of modal mass contributing to the Coriolis effect. This factor is intrinsically determined by vibratory-mode geometry and always constrained to $\kappa_0 \leq 1$ (Supplementary Note 1). Consequently, for small rotation rates Ω , the strength of the Coriolis coupling, $2\kappa_0\Omega$, is weak. The resulting physical modulations (e.g. amplitude or frequency changes) are easily blurred by the inherently stronger Brownian noise of microscale resonators in chip-scale CVGs relative to macroscale HRGs. Whether and how the Coriolis effect itself can be enhanced beyond this sensitivity limit imposed by the Coriolis factor, $\kappa_0 \leq 1$, remains an unresolved outstanding question. Resolving this barrier could enable HRG-level performance in chip-scale CVGs and unlock transformative applications."

(2) Accordingly, the **Abstract** has been revised as:

"Gyroscopes, as fundamental inertial sensors, are crucial for rotation measurements in the consumer electronics, automotive, and aerospace industries, with the most widely used kind relying on the Coriolis effect¹⁻⁶. The chip-scale Coriolis vibratory gyroscopes (CVGs) show reduced size, weight, and cost^{1,2}, but remain far lower performance than traditional macroscale CVGs³⁻⁶, as the weak intrinsic Coriolis factor sets a fundamental limit on scaling the sensitivity against the louder Brownian noise in microchips compared to the macroscale ones. Here, to overcome this physical limit, for the first time, we propose and experimentally demonstrate the use of third-order singularities lying within cusp catastrophes in the phase-tracked oscillations of an on-chip CVG to facilitate a cubic-root scaling of the Coriolis-effect-induced frequency modulation. Employing this effect, we achieve a three-order-of-magnitude enhancement in the Coriolis factor, yielding a 253-fold improvement in signal-to-noise ratio and a 297-fold increase in precision. Moreover, the cusp singularity enables a

previously unattainable ultrasensitive phase-modulated sublinear measurement, achieving a world-record signal-to-noise ratio performance for silicon-chip gyroscopes. These findings not only provide revolutionary advancements in gyroscope technologies, by filling the gap in observing and controlling the singularity-enhanced Coriolis effect, but also shed new light on other ultrasensitive sensing applications.”

(3) New thermomechanical-noise analysis introduced in the revised Supplementary Information.

To assess how thermomechanical noise affects the singularity-enhanced frequency- and phase-modulated operations, we have added Supplementary Note 11: “Thermomechanical noise model for singularity-enhanced frequency and phase modulations” in the revised Supplementary Information. This phasor analysis shows that thermomechanical noise is not amplified at the cusp singularities relative to conventional frequency-modulated theory, and the Brownian-motion induced phase (frequency) fluctuations enter additively into the phase (frequency) outputs, without singularity-induced gain.

(4) Added discussion in the main text (last paragraph, p. 14).

We have added a discussion about the thermomechanical noise in the singularity-enhanced operations:

“Crucially, we find that the Brownian-motion–induced phase (frequency) noise enters additively into the PM (FM) outputs, without singularity-induced gain (Supplementary Note 11). By contrast, the singular Coriolis responses follow boosted cubic-root responsivities, enabling signal-to-noise ratios that exceed the fundamental limit of the conventional CVG theory. As a result, the singularity-mediated PM readout surpasses the resonator’s theoretical performance bounds in conventional operation, where bias instability and angle random walk (ARW) are theoretically $> 0.32^\circ/\text{h}$ and $> 0.018^\circ/\sqrt{\text{h}}$, respectively (Supplementary Note 12). These results establish singularity-mediated PM as a fundamentally superior transduction mechanism.”

(5) We have included a very important reference about the phasor analysis and the noise theory of the conventional FM gyroscope in the revised manuscript:

30. M. Kline, *Frequency modulated gyroscopes*. (University of California, Berkeley, CA, 2015).

Comment 2.3: "This coefficient measures the proportion of modal mass contributing to the Coriolis effect and is intrinsically determined by 25 vibratory-mode geometry, always constrained to ≤ 1 ".

As a further confirmation of the points above, this is valid also for macroscale devices!

Response: Indeed, the constraint $\kappa_0 \leq 1$ applies equally to macroscale devices. In fact, low Coriolis efficiency has also posed challenges for the early development of high-performance HRGs. As reviewed in the well-known article "*The Hemispherical Resonator Gyro: From Wineglass to the Planets,*" early aluminum HRGs did not exhibit outstanding performance. It was the transition to fused silica, enabling quality factors exceeding 10 million, that allowed HRGs to achieve their superior results. Within the framework of conventional CVG theory, realizing a high-performance microscale silicon-chip gyroscope is even more demanding, because the more pronounced thermomechanical noise at the microscale makes attaining a sufficiently high SNR significantly harder.

Comment 2.4: The baseline for the authors study, which is a QFM, is known to be not good for long term navigation purposes due to the drifts of the frequencies. Besides, note that in a QFM both the frequencies change as $w_0 + k_0 \cdot \Omega$.

Response: We agree that pure QFM, particularly single-channel QFM (the QFM with only a single CW or CCW operation is initiated), is not ideal for high-precision rotation sensing because of limited long-term stability. That said, QFM remains one of the most elegant and fundamental operations realized in MEMS resonators: a clean, circular traveling-wave oscillation. Its simplicity and physical clarity make it an excellent platform for fundamental studies.

For example, rotating QFM reveals the angular-gain factor explicitly as the scale factor of frequency modulation, linking two seemingly different phenomena—the Coriolis effect and the acoustic rotational Doppler effect [Science Advances 7, eabd6705 (2021)]. The QFM also serves as a compelling acoustic analog of spin and its rotation-induced frequency modulation is closely related to the acoustic analog of the Zeeman effect [Science 343, 516 (2014)]. Beyond the singularity physics demonstrated here, we

believe QFM would be a powerful testbed for exploring many other physical effects [e.g. Science 343, 516 (2014); Nature Physics 16, 571–578 (2020)] and for engineering other practical devices.

Because of its simplicity and purity in physics, we chose QFM to demonstrate the singularity-enhanced Coriolis effect, aiming to make the underlying ideas more easily understood by a broader audience.

Thank you also for the reminder regarding frequency shifts in QFM. We apologize for the earlier ambiguity. In the original manuscript, we did **not** intend to suggest that the two standing-wave modes shift as $\omega + \kappa_0\Omega$ and $\omega - \kappa_0\Omega$ simultaneously. Our intended meaning was that the oscillation frequency shifts as $\omega + \kappa_0\Omega$ for a CW operation, or as $\omega - \kappa_0\Omega$ for a CCW operation. We have revised the text accordingly to avoid misunderstanding.

Changes made:

In lines 9-13, page 4 of the manuscript, the original confusing statement: “In this operation, modes 1 and 2 are simultaneously excited into equal amplitude and quadrature phase difference, creating a circular travelling-wave (TW) mode, depicted by the circular orbit of the proof mass in Fig. 1a, exhibiting a resonant frequency of $\omega \pm \kappa_0\Omega$ ” has been revised as:

“In this operation, modes 1 and 2 are simultaneously excited into equal amplitude and quadrature phase difference, creating a circular travelling-wave (TW) mode, depicted by the circular orbit of the proof mass in Fig. 1a, exhibiting a resonant frequency of $\omega + \kappa_0\Omega$ or $\omega - \kappa_0\Omega$ for clockwise (CW) or counterclockwise (CCW) operations, respectively.”

Comment 2.5: The introduced idea is new but... A starting situation with a bias stability of 800 °/hr is not at all the state of the art (even in my pocket I have better gyros in the mobile phone). There are QFM and LFM reaching bias stability of 1 °/hr or sub °/hr. Additionally, there are AM gyroscopes reaching 0.00x °/hr. So, the 2x-fold improvement shown by the authors is true only against a very poor starting sensor.

As such, in my opinion the paper cannot be accepted, because final results lie well

below state of the art performance of microgyroscopes.

Response: We sincerely appreciate your positive assessment of our work’s novelty. In this paper, we report, for the first time, cusp-singularity enhancement of the Coriolis effect. By operating near singularities within cusp catastrophes, we generate ultrasensitive frequency- and phase-modulated responses and ultrahigh SNR with sublinear scaling of the Coriolis outputs, thereby surpassing the fundamental sensitivity and SNR limits of conventional Coriolis gyroscopes.

2.5.1 QFM as the baseline for singularity enhancement

To clarify, we did not intend to use the $800^\circ/\text{h}$ QFM operation as the state-of-the-art gyroscope performance baseline. In our study, the singular Coriolis effect enables a *singularity-mediated phase-modulated (PM)* operation that runs concurrently with the singularity-enhanced frequency-modulated (FM) operation and outperforms FM (see Section “Singularity-mediated phase modulation for rotation measurement” in the main text). The experimental bias instability ($0.035^\circ/\text{h}$) and ARW ($0.00036^\circ/\sqrt{\text{h}}$) of the X_1 -mediated PM operation are compared with the **theoretical limits** of conventional AM (see p. 14, lines 30) and with **state-of-the-art silicon-chip gyroscopes** developed by Boeing, Honeywell, Silicon Sensing, and other top players in the field (see Extended Data Table 1). The ARW of $0.00036^\circ/\sqrt{\text{h}}$ is also comparable to those of HRGs (see Fig. 4e). If we had adopted QFM as the performance baseline, we would have claimed enhancement factors of **22,874** in bias instability ($800.6^\circ/\text{h}$ to $0.035^\circ/\text{h}$) and **45,000** in ARW ($16.2^\circ/\sqrt{\text{h}}$ to $0.00036^\circ/\sqrt{\text{h}}$), but we intentionally did not make such claims. We realize that the original statement in the Discussion and Conclusions, “Moreover, leveraging the enhanced Coriolis effect, we have showcased a chip-scale CVG achieving HRG-comparable, strategic-grade ARW”, was misleading and led to this misunderstanding. In the revised manuscript, we clarify that it is the **phase-modulated output** of the enhanced Coriolis effect that demonstrates the strategic-grade ARW.

Respectfully, we emphasize that the central contribution of this work is **not** the engineering of a high-performance microgyroscope per se, although we achieve an **unexpected** world-record noise performance for silicon-chip gyroscopes, but rather the **new operating principle** and its huge potential for sensitivity enhancement. To make the singularity-enhanced Coriolis effect as transparent as possible, we chose single-channel QFM as our demonstration platform because it is simple, intuitive, and conceptually clean, enabling the underlying physics to be easily understood by a

broader audience.

In our opinion, the relatively modest performance of standard single-channel QFM does not diminish the large enhancement factors we report in responsivity, SNR, and precision. The $800^\circ/\text{h}$ stability reflects the true performance of our resonator in single-channel QFM; it was not intentionally degraded. For reference, a pure single-channel QFM microgyroscope developed by the University of California, Berkeley, shows a stability of $>1000^\circ/\text{h}$ [M. Kline, Frequency Modulated Gyroscopes, Ph.D. thesis, University of California, Berkeley, (2015). M. Kline, et al., Quadrature FM gyroscope, MEMS (2013) pp. 604–608], which is comparable to our result. Moreover, the **singularity-enhanced FM** and **standard QFM** measurements were conducted under **identical** environmental and device conditions, subject to nearly the same internal and external noise sources. The added coupling required for singularity-enhanced FM can even introduce **additional** error sources, as evidenced by Fig. 3d (copied below for your convenience), where the singularity-enhanced FM exhibits **worse** noise and error in frequency outputs than standard QFM. Therefore, the enhancement factors obtained by comparing singularity-enhanced FM with standard QFM are **meaningful** and directly reflect the power of the cusp singularity in sensing enhancement.

Fig. 3d. Allan deviations of the zero-rotation bias frequency signals for singularity-enhanced frequency-modulated operations and standard QFM operation. The singularity-enhanced frequency-modulated operations show worse noise and errors in the frequency output signals than the standard QFM operation.

We acknowledge that various technical refinements can raise QFM performance to higher, tactical-grade levels, e.g., space-domain differential outputs [dual-channel QFM; M. Kline *et al.*, MEMS 2013, pp. 604–608], time-domain differential outputs [indexed FM; B. Eminoglu *et al.*, MEMS 2016, pp. 954–957], and Lissajous FM [frequency mismatch

is introduced, M. Kline, Ph.D. thesis, 2015; I. I. Izyumin *et al.*, MEMS 2015, pp. 33–36]. However, we believe that these engineering improvements are orthogonal to our principal innovation and not required to demonstrate the cusp-singularity enhancement of the Coriolis effect. Differential methods can also be applied to our singularity-enhanced operations and may further improve performance, but we consider these directions appropriate for **future** explorations because they are irrelevant to the innovation of this study.

2.5.2 Singularity-mediated phase-modulated operation

Here, we would like to present a kind reminder that, beyond enhanced FM, the cusp singularity also facilitates a **phase-modulated (PM)** readout, using the **relative phase** between the two standing-wave modes as an ultrasensitive rotation output (see the Section “Singularity-mediated phase modulation for rotation measurement”). Unlike standard QFM, where the relative phase is constant, cusp singularities produce **cubic-root sublinear modulations** of the relative phase at small angular velocities. Operating at the X_1 cusp singularity, we obtain a bias instability of $0.035^\circ/\text{h}$ and an ARW of $0.00036^\circ/\sqrt{\text{h}}$.

The singularity-mediated PM outperforms the singularity-enhanced FM for two main reasons. **First**, unlike frequency readout, the **relative phase** is inherently robust to resonant-frequency fluctuations, enabling significantly higher stability (see Supplementary Note 9). **Second**, phase noise is the **time integral** of frequency noise. By applying the Fourier transformation, the phase noise equals the frequency noise times $1/i\omega$ in the frequency domain. Consequently, the phase-measurement noise spectral density is suppressed by $1/\omega_0^2$ relative to frequency detection (Supplementary Note 11), yielding substantially lower noise.

2.5.3 Comparison with state-of-the-art silicon-chip microgyroscopes

With respect, we disagree that our final results fall well below the state of the art. Extended Data Table 1 (copied below for your convenience) presents a detailed comparison with leading silicon-chip microgyroscopes from Boeing, Honeywell, Silicon Sensing, and other top players in the field. Our **singularity-mediated PM** operation achieves **strategic-grade** ARW of $0.00036^\circ/\sqrt{\text{h}}$, outperforming current cutting-edge silicon-chip gyroscopes by nearly **an order of magnitude**.

As discussed in Response 2.1.4, the principal bottleneck for silicon-chip microgyroscopes is the **thermomechanical (Brownian) noise**, so ARW (i.e., SNR)

stands as the most critical challenge. The singularity-enhanced Coriolis effect breaks the conventional CVG SNR limit by greatly **boosting** the Coriolis response at small angular velocities, enabling ARW that exceeds the theoretical bounds of the standard theory.

Extended Data Table 1. Comparison to state-of-the-art high-performance silicon-chip Coriolis vibratory gyroscopes.

	Config.	Struct.	Resonant Freq., Detuning (if any)	Q	Intrinsic Coriolis factor	Proof mass	Drive Ampl.	ARW ($^{\circ}/\sqrt{h}$)	Bias stability ($^{\circ}/h$)	Year
Boeing, JPL [39]	AM	Disk	14 kHz	80k	0.8	1.1 mg	1.8 μm	0.0021	0.012 (Allan)	2014
University of California, Irvine [40]	AM	Quad-mass tuning fork	1.6 kHz	1.8M	0.87	8 mg	2.5 μm	0.015	0.09 (Allan)	2016
Honeywell [41]	AM	Dual-mass tuning fork	--, >700 Hz	--	>0.9	--	--	0.003	0.009 (Allan) 0.05 ($1\sigma@10s$)	2019
Northrop Grumman LITEF GmbH [42]	AM	Dual-mass tuning fork	--	--	>0.9	--	--	0.02	0.007 (Allan) 0.3 ($1\sigma@10s$)	2019
Politecnico di Milano [43]	AM	Dual-mass tuning fork	25 kHz, 100-200 Hz	25k-50k	>0.9	3.5 μg	9 μm	0.004	0.02 (Allan)	2021
Silicon Sensing CRS39A	AM	Ring	14 kHz	--	0.8	--	--	0.004	0.03 (Allan)	2021
NUDT [44]	AM	Disk	4.2k	570k	0.85	1.8 mg	2 μm	0.003	0.015 (Allan, standard), 0.003 (Allan, mode reversal&deflection) 0.08 ($1\sigma@10s$, mode reversal&deflection)	2025
This work	Singularity enhanced	Disk	40.4kHz	112k	0.588	60 μg	0.5 μm	0.00036	0.035 (Allan)	2025

We also demonstrate **near-inertial-grade** bias instability (Allan deviation) of $0.035^{\circ}/h$. While not a world record, it is highly competitive for silicon-chip devices. Numerous techniques can further improve bias stability—for example, our team members demonstrate that mode-reversal with deflection control can reduce bias instability from $0.015^{\circ}/h$ to $0.003^{\circ}/h$ in a standard AM microgyroscope [ref.44 Chen et al., $0.003^{\circ}/h$ bias instability of honeycomb disk resonator gyroscope achieved by mode reversal combined mode deflection control method, *Microsystems & Nanoengineering* (2025)11:152], but such methods **do not** improve SNR (ARW).

Indeed, some AM silicon-chip gyroscopes have demonstrated $0.00x^\circ/h$ Allan deviation bias instability. For example, the tuning-fork microgyroscopes from Honeywell [ref. 41] and Northrop Grumman LITEF GmbH [ref. 42], as well as the disk-resonator gyroscope developed by our team members [ref. 44]. These represent the state of the art in silicon-chip gyroscopes. Nevertheless, they still suffer from low SNR, and **a stable yet noisy gyroscope still requires SNR improvements**. The standard deviation bias stability at an averaging time of 10s ($1\sigma@10s$) is a more informative metric to characterize these devices, as it reflects both stability and SNR. The $1\sigma@10s$ bias stabilities of these devices, which are provided in Extended Data Table 1, fall in the range of $0.05\text{--}0.3^\circ/h$. To illustrate this, we provide the Allan deviation data of [ref. 44] in Fig. R3. While the minimum of the Allan-deviation curve (Allan bias instability, BI) reaches $0.003^\circ/h$, the $1\sigma@10s$ standard-deviation bias stability, taken at a 10s integration (sampling) time, is notably worse at $0.08^\circ/h$. **The silicon-chip microgyroscopes desperately call for a higher SNR**, which would shift the Allan-deviation curve downward and generate better $1\sigma@10s$ bias stability.

Fig. R3. Allan deviations of the disk resonator gyroscope employing mode-reversal with deflection control [ref. 44]. The Allan deviation bias instability (BI) can reach $\sim 0.003^\circ/h$, the $1\sigma@10s$ standard deviation bias stability is about $0.08^\circ/h$, and the ARW is approximately $0.003^\circ/\sqrt{h}$.

Our study breaks the conventional SNR limit and demonstrates strong potential for a strategic-grade silicon-chip gyroscope. The reported bias instability is based on an **early-stage, unoptimized prototype** and is expected to improve further. Specifically: (1) the intrinsic Coriolis factor of our disk resonator is 0.588; selecting appropriate $n=2$

wineglass modes could raise it above 0.8, yielding an immediate gain; (2) differential, singularity-enhanced operation, either simultaneous or alternating CW/CCW, typically can improve bias stability by several-fold; and (3) in this prototype, stiffness coupling is applied via DC voltage; its fluctuations can degrade stability. employing **laser trimming** or other **mechanical tuning** techniques could provide more stable coupling. With these future refinements, we expect substantially higher stability and precision.

2.5.4 Advancements in singularity physics

This work also represents a significant advance in **singularity-enhanced sensing**, a rapidly growing field initiated by the theoretical work [J. Wiersig, Physical Review Letters 112, 203901 (2014)]. Milestone experimental developments in this field are summarized in Extended Data Table 2 (copied below for your convenience).

Extended Data Table 2. Comparison to previous experiments of singularity-enhanced sensing.

Reference	Singularity type (order)	Degrees of freedom	Enhancement object	Enhancements			Year
				Responsivity	SNR	Precision	
Chen et.al. Nature 548, 192 (2017).	Exceptional point (2)	2	Nanoparticle sensing	2.5×	--	--	2017
Hodaei et.al. Nature 548, 187 (2017).	Exceptional point (3)	3	Joule heat sensing	23×	--	--	2017
Hokmabadi et.al. Nature 576, 70 (2019).	Exceptional point (2)	2	Sagnac effect	20×	--	--	2019
Lai et.al. Nature 576, 65 (2019).	Exceptional point (2)	2	Sagnac effect	4×	Ineffective	Ineffective	2019
Kononchuk et.al. Sci. Adv. 7, eabg8118 (2021).	Wigner's cusp anomaly (2)	2	Acceleration sensing	60×	8.4×	7.8×	2021
Kononchuk et.al. Nature 607, 697 (2022).	Exceptional point (2)	2	Acceleration sensing	10×	3×	~3×	2022
Suntharalingam et.al. Nat. Commun. 14, 5515 (2023).	Nonlinear exceptional point (2)	2	Voltage sensing	100×	>150×	>4.5×	2023
Xu et.al. Nat. Nanotechnol. 19, 1472 (2024).	Exceptional point (2)	2	Nanometrology	86×	5×	~8×	2024
Ruan et.al. Nat. Photon. 19, 109 (2025).	Exceptional point (2)	2	Magneto-optical effect	10×	3×	--	2024
This work	Cusp singularity (3)	2	Coriolis effect	1010×	253×	297×	2025

We show, for the first time, that cusp-catastrophe-supported singularities can deliver giant sensing enhancements: a three-order-of-magnitude increase in sensitivity, a 253-fold improvement in SNR, and a 297-fold increase in precision relative to the conventional sensing method. As explained in Response 2.5.1, absolute performance evaluations of the single-channel FM operations are meaningless because they all suffer from resonant-frequency fluctuations; however, relative comparisons are valid since the

singularity-enhanced FM and QFM were tested under identical conditions. These enhancements, among the highest reported in recent singularity-enhancement experiments, together with the ultrasensitive, cusp-singularity-enabled phase-modulated operation demonstrated here, advance the physics of singularity-enhanced sensing.

2.5.5 Contribution summary

Finally, we would like to summarize the threefold contributions of this study for your reference:

(1) For the first time, we resolve the previously open question of whether and how the Coriolis effect can be enhanced. We demonstrate cube-root scaling of the Coriolis effect by operating near third-order singularities within cusp catastrophes, an advance beyond the conventional view that the Coriolis output is always proportional to the angular rotation. Exploiting this singular Coriolis effect allows us to surpass the intrinsic Coriolis-factor limit on Coriolis vibratory gyroscope sensitivity, yielding substantial gains in sensitivity, precision, and SNR.

(2) The cusp singularity further enables a novel phase-modulated operating mode that achieves, to our knowledge, world-record strategic-grade SNR for silicon-chip gyroscopes. Our findings challenge the traditional view that miniaturized gyroscopes necessarily suffer degraded SNR and could have immediate implications for the inertial-sensing industry. This breakthrough may enable revolutionary technologies such as high-performance nanoscale gyroscopes and compact, low-cost inertial north finders.

(3) We show, for the first time, that cusp-catastrophe-supported singularities can deliver such large gains in responsivity, SNR, and precision, advancing the physics of singularity-enhanced sensing. The resulting improvements, among the highest reported in recent singularity-enhancement experiments, together with the ultrasensitive, cusp-singularity-enabled phase-modulated operation demonstrated here, suggest a general strategy for surpassing classical sensitivity limits across diverse domains, potentially including healthcare diagnostics, seismology, environmental monitoring, and even gravitational-wave detection, where extreme sensitivity is critical.

We would like to express our sincere gratitude for your valuable feedback, which has significantly strengthened the clarity and quality of the paper. We have incorporated additional analyses, thorough revisions, and clarifications, and we hope these changes

and responses address all of your concerns.

Changes made:

(1) In lines 16-19, page 12 of the manuscript, we have made an introduction of the singularity-mediated phase modulation:

“Furthermore, we find that when the PhT system operates near the cusp singularities, the relative phase ϑ can be a superior metric for rotation readout than the PhT frequency ω_T . This enables a novel phase-modulated (PM) gyroscope architecture that achieves strategic-grade noise performance in silicon chips.”

(2) In lines 18-23, page 14 of the manuscript, we have made an additional explanation of why the singularity-mediated phase modulation can provide higher SNR and stability than the singularity-enhanced frequency modulation:

“This PM operation delivers over two orders of magnitude improvement in both stability and SNR relative to FM operation. First, unlike ω_T , the relative phase ϑ is inherently unaffected by resonant-frequency fluctuations (Supplementary Note 9), thereby enabling significantly higher measurement stability. Second, because frequency noise is integrated into phase noise, the corresponding spectral density acquires a $1/\omega_0^2$ suppression factor (Supplementary Note 11), yielding much lower noise in PM readout compared with FM detection.”

(3) In lines 11, page 15 of the manuscript, the original representation of “leveraging the enhanced Coriolis effect, we have showcased a chip-scale CVG achieving HRG-comparable, strategic grade ARW” as “leveraging the PM output of the enhanced Coriolis effect, we have showcased a chip-scale CVG achieving HRG-comparable, strategic grade ARW” to avoid confusing.

(4) At the end of the paper, we have made a statement that introducing differential architectures to the singularity-enhanced sensing may further enhance the stability of the gyroscope:

“At present, the demonstrated cusp-singularity-enhanced measurements employ a single-channel, non-self-calibrated configuration. Future implementations that incorporate differential architectures^{29,30,50} could further improve bias stability.”

(5) We have also included two references that describe the differential architectures that are applicable to the singularity-enhanced sensing:

30. M. Kline, Frequency modulated gyroscopes. (University of California, Berkeley, CA, 2015).

50. B. Eminoglu, Y.-C. Yeh, I. I. Izyumin, I. Nacita, M. Wireman, A. Reinelt, B. E. Boser, Comparison of long-term stability of AM versus FM gyroscopes. In Proc. 2016 IEEE 29th International Conference on Micro Electro Mechanical Systems (MEMS) 954–957 (IEEE, 2016).

(6) We have updated Extended Data Table 1 to include the $1\sigma@10s$ bias stabilities of some state-of-the-art silicon-chip gyroscopes.

Response to Referees

Referees' comments:

Referee #2 (Remarks to the Author):

Comment 1.0: Thank you for answering the different points, and for providing extensive explanations.

The work can now be accepted.

Response: We sincerely thank the Referee for the constructive comments on our paper and are deeply grateful for your valuable contributions, which have greatly helped us improve our work.